# Specification of diverse cell types during early neurogenesis of the mouse cerebellum

John W Wizeman[1†], Qiuxia Guo[1†], Elliott M Wilion[2], James YH Li[1,3]*

[1]Department of Genetics and Genome Sciences, School of Medicine, University of Connecticut, Farmington, United States; [2]University of Connecticut, Storrs, United States; [3]Institute for Systems Genomics, University of Connecticut, Farmington, United States

**Abstract** We applied single-cell RNA sequencing to profile genome-wide gene expression in about 9400 individual cerebellar cells from the mouse embryo at embryonic day 13.5. Reiterative clustering identified the major cerebellar cell types and subpopulations of different lineages. Through pseudotemporal ordering to reconstruct developmental trajectories, we identified novel transcriptional programs controlling cell fate specification of populations arising from the ventricular zone and the rhombic lip, two distinct germinal zones of the embryonic cerebellum. Together, our data revealed cell-specific markers for studying the cerebellum, gene-expression cascades underlying cell fate specification, and a number of previously unknown subpopulations that may play an integral role in the formation and function of the cerebellum. Our findings will facilitate new discovery by providing insights into the molecular and cell type diversity in the developing cerebellum.

DOI: https://doi.org/10.7554/eLife.42388.001

*For correspondence:
jali@uchc.edu

†These authors contributed equally to this work

Competing interests: The authors declare that no competing interests exist.

## Introduction

The cerebellum plays an important role in cognitive processing and sensory discrimination, in addition to its better-known function in motor coordination (*Ito, 2008*). Development of the cerebellum is an excellent model for studying neurogenesis and assembly of neural circuits because the cerebellum has a relatively simple and highly stereotyped cytoarchitecture (*Altman and Bayer, 1997*; *Sillitoe and Joyner, 2007*). The entire cerebellum is believed to be derived from the alar plate of rhombomere 1, the anterior-most segment of the metencephalon (*Millet et al., 1996*; *Sunmonu et al., 2011*; *Wingate and Hatten, 1999*; *Zervas et al., 2004*). The cerebellar neurons that use gamma-aminobutyric acid (GABA) or glutamate as transmitters differentially arise from the cerebellar ventricular zone (VZ) and rhombic lip (RL), two spatially distinct germinal zones of the cerebellar anlage (*Ben-Arie et al., 1997*; *Hoshino et al., 2005*; *Pascual et al., 2007*; *Wang et al., 2005*). The cerebellar VZ also produces various glial cells, including Bergmann glia, oligodendrocytes, and astrocytes (*Grimaldi et al., 2009*; *Sudarov et al., 2011*). In addition, different types of GABAergic and glutamatergic neurons are generated in sequential and temporally restricted phases (*Hashimoto and Mikoshiba, 2003*; *Machold and Fishell, 2005*; *Sudarov et al., 2011*; *Wang et al., 2005*). These observations have provided a basic framework for our understanding of cerebellar development. However, it is evident that cerebellar cell types have a previously unappreciated heterogeneity. For example, up to 50 distinct Purkinje cell (PC) clusters are present in the mouse cerebellum at the late gestation stages; these PC clusters are subsequently transformed into longitudinal stripes along the mediolateral axis (*Dastjerdi et al., 2012*; *Sillitoe and Joyner, 2007*). The exact extent of cell heterogeneity, and how the diversity arises in the developing cerebellum, are not

completely clear. In particular, we lack cell-specific markers for cerebellar cell populations during early neurogenesis. This missing information is crucial for understanding the molecular mechanisms underlying cell fate specification of early cerebellar development.

In this study, we sought to define the molecular features and developmental trajectories of different cerebellar cell types during early neurogenesis of the developing cerebellum. We performed single-cell RNA sequencing (scRNAseq) of the mouse cerebellum at embryonic day (E) 13.5, when active proliferation and cell fate determination occurs. We identified at least 22 groups of neural and non-neural cells. We determined specific markers that can be used to identify these cell groups and to infer their developmental trajectories. We uncovered a novel signaling center that may play an important role in the formation of the cerebellar vermis, and five putative subtypes of PC precursors. Using detailed expression and genetic fate-mapping studies, we not only characterized the spatial distribution of the cell groups identified by scRNAseq, but also resolved their developmental origins.

## Results

### Single-cell transcriptomics identifies major cell groups in E13.5 mouse cerebella

To characterize cellular heterogeneity in the developing cerebellum, we used the Chromium system (10x Genomics) to profile the transcriptome of nearly 10,000 single cells from E13.5 mouse cerebella. After filtering outlier cells and genes, 9,326 cells and 15,823 genes were kept for subsequent analysis. By applying the Seurat pipeline (*Butler et al., 2018*; *Satija et al., 2015*), we identified 19 major cell groups and their defining molecular features (*Figure 1A*, and *Supplementary file 1*). Based on the combination of known markers, we grouped these cell clusters into four categories: 1) VZ neural progenitor cells (NPCs) that express *Hes1* and *Hex5* (*Kageyama et al., 2008*); 2) GABAergic neurons and their precursors that express *Lhx1* and *Lhx5* (*Morales and Hatten, 2006*; *Zhao et al., 2007*); 3) glutamatergic neurons and their precursors that express *Atoh1* and *Barhl1* (*Ben-Arie et al., 1997*; *Li et al., 2004a*); 4) non-neural cells, including endothelial cells, pericytes, and erythrocytes (*Figure 1B*). To evaluate the vigor of our results, we repeated cell clustering with subsets of the data (random sampling of 70, 50, or 30% of total cells; n = 3 for each sampling). Although the consistency that a given cell was classified to a certain group decreased as the number of cells decreased, the identified cell groups and their proportions were highly reproducible between the original and downsampled datasets (*Figure 1C and D*). These results demonstrate the robustness of our initial cell clustering.

### Novel signaling centers within the cerebellar anlage

Refined clustering of presumptive NPCs (cluster 3, 5, 6, and eight in *Figure 1A*) revealed four cell groups (*Figure 2A*). We performed differential expression analysis to identify feature genes of each cell group (*Supplementary file 1*). Through functional and pathway enrichment analysis (*Huang et al., 2007*), we detected no significant enrichment in group one feature genes, whereas group two genes were enriched for those involved in proteinaceous extracellular matrix and cell differentiation (*Supplementary file 2*). The feature genes of groups 3 and 4 encode molecules that are significantly enriched in the Wnt signaling pathway, including *Fzd9/10*, *Dkk2*, *Rspo1/3*, *Wls*, *Wif1*, and *Wnt1/3a/9a* (*Figure 2B* and *supplementary file 2*). In addition, group 4 cells express *Fgf17*, *Calb2,* and other genes that are absent from group 3 (*Figure 2B* and *Supplementary file 1*).

We next used immunohistochemistry (IHC) and in situ hybridization (ISH) to relate the four NPC cell groups to their endogenous position in the cerebellar anlage. We found that feature genes of group 1–3 are expressed in distinct domains of the VZ: group 1's genes are in the anterior; group 2's in the posterior; and group 3's in the posterior-most region of the VZ opposing the RL (*Figure 2C* and *Figure 2—figure supplement 1A–H*). The group 3's position resembles the so-called C1 domain defined previously (*Chizhikov et al., 2006*). Group four was related to a longitudinal column of cells along the midline of the cerebellar anlage (*Figure 2C* and *Figure 2—figure supplement 1I–O*). We confirmed the presence of Fgf17 protein surrounding the cerebellar midline cells that express *Fgf17* mRNA and Calb2 protein at E13.5 (*Figure 2D* and *Figure 2—figure supplement 2A,B*). Phosphorylated extracellular signal-regulated kinases (ERKs) were detected in

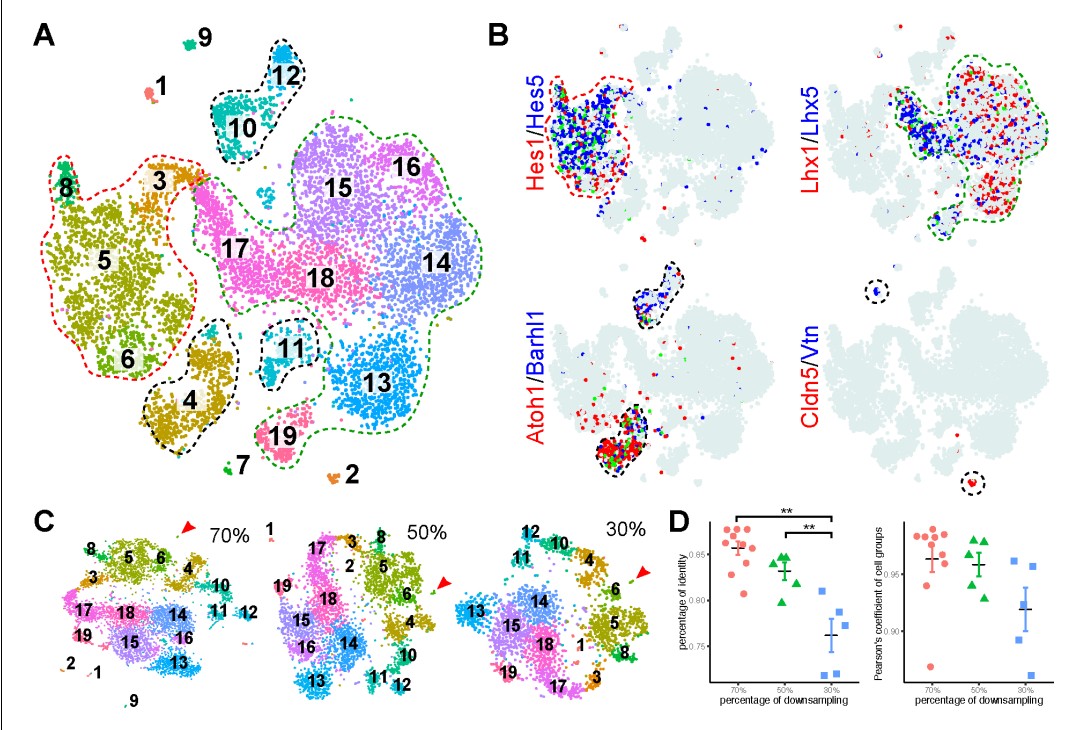

**Figure 1.** Identification of major cell types in E13.5 mouse cerebella by scRNAseq. (A) Visualization of 19 classes of cells using t-distributed stochastic neighbor embedding (tSNE). Each dot represents a cell, similar cells are grouped and shown in colors. The colored dashed lines denote the major cell types. (B) Expression of known markers is shown as laid out in A (red and blue, expression of individual markers; green, co-expression; azure, no expression). The marker-expressing cell groups are outlined by dashed lines. (C) tSNE plotting of clustering of randomly downsampled datasets in 70%, 50% and 30% of the original cells. Note that almost the same clusters indicated by number and color are found in the smaller datasets, except for the small cluster shown by the arrowhead. (D) Scatter plots showing the percentage of identity (left, **p < 0.01, one-way ANOVA with post-hoc Tukey HSD test) and Pearson's coefficient of the cell group proportion (right).

DOI: https://doi.org/10.7554/eLife.42388.002

association with the Fgf17 immunoreactivity, indicating the activation of the Fgf-ERK signaling pathway at the midline of the cerebellar anlage (*Figure 2E*). The *Fgf17*-expressing midline cells were surrounded not only by a layer of Mki67+ cells (proliferating), but also numerous EdU+/Mki67- cells, which represented cells that had exited cell cycle within 24 hr following EdU pulse-chase labeling (*Figure 2F*). Remarkably, although most Sox2+ cells in the VZ were positive for Mki67, the midline cells displayed robust Sox2 immunoreactivity, but were negative for Mki67 and EdU, suggesting that they are quiescent progenitors (*Figure 2F*). pERK immunoreactivity was specifically absent from the C1 domain (*Figure 2G*), indicating the lack of Fgf-ERK signaling at this region. Using a BAT-gal reporter transgene (*Maretto et al., 2003*), we confirmed Wnt/ß-catenin signaling activity in the C1 domain and midline area between E12.5 and E14.5 (*Figure 2H*). These observations are in agreement with previous reports (*Lancaster et al., 2011*; *Selvadurai and Mason, 2011*). Our data suggest that the cerebellar midline cells and C1 cells are organizing centers through Wnt/ß-catenin signaling. In addition to Wnt/ß-catenin activity, the midline cells pattern the developing vermis via Fgf-ERK signaling. We named the cerebellar midline cells as the midline organizer (MidO).

It has been shown that Fgf8 induces specialized roof plate (RP) cells called the isthmic node, which express *Gdf7* and *Wnt1* at the isthmus, in chick embryos (*Alexandre, 2003*). Cells from the isthmic node subsequently populate the entire length of the cerebellar midline (*Alexandre, 2003*; *Louvi et al., 2003*). Genetic fate mapping in mice demonstrates that *Gdf7*-lineage cells form a longitudinal column along the cerebellar midline (*Cheng et al., 2012*). Using a *Wnt1-creER* transgene (*Zervas et al., 2004*), we showed that the descendants of *Wnt1*-expressing cells labeled at E8.5 contributed to the midline (*Figure 2—figure supplement 2C and D*), and they express Lmx1a and Calb2 at E16.5 (*Figure 2I*). Notably, although MidO cells exhibited Lmx1a immunoreactivity at least

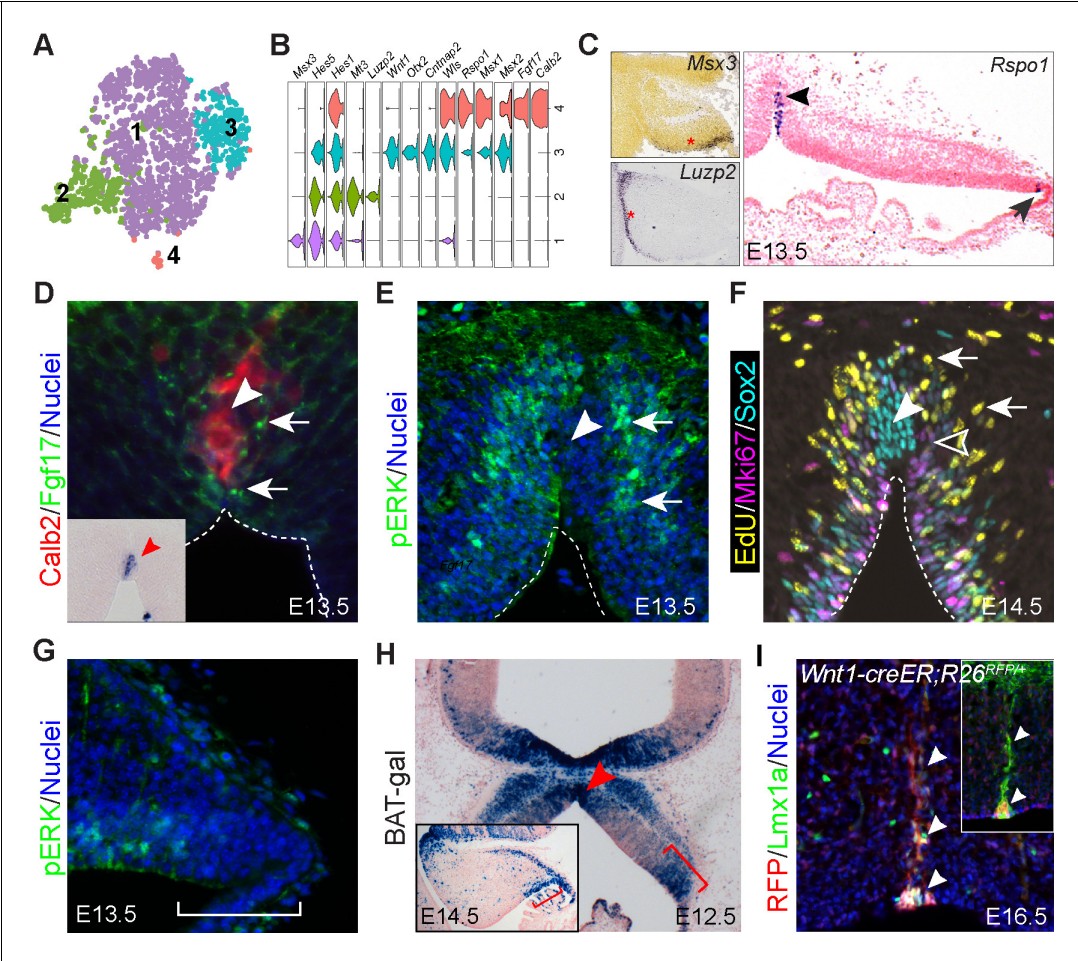

**Figure 2.** Identification of signaling centers in the cerebellar primordium. (**A**) tSNE showing the partition of progenitors in the cerebellar VZ. (**B**) Violin plots showing cell-specific markers. (**C–G**) ISH (**C**) and IHC (**D–G**) on coronal sections of cerebella at the indicated embryonic stage. The arrowhead and arrow in C indicate the MidO (group 4) and C1 (group 3), respectively; arrows denote Fgf17 (**D**), pERK signals (**E**), and EdU$^+$/Mki67$^-$ newly born cells (**F**); the empty arrowhead shows EdU$^-$/Mki67$^+$ cells; the arrowhead denotes the MidO; the bracket in G demarcates the C1 domain. Inset in D showing ISH for *Fgf17* on a section adjacent to D. (**H**) X-gal histochemistry on a coronal section of E12.5 cerebellum carrying the BAT-gal transgene. The inset shows a sagittal section of E14.5 cerebellum; the brackets demarcate C1; the arrowhead indicates the MidO. (**I**) IHC on a coronal cerebellar section of E16.5 *Wnt1-creER;R26$^{RFP/+}$* embryo that received tamoxifen at E8.5. The arrowheads indicate colocalization of Lmx1a and tdTomato red fluorescent protein (RFP). The inset shows colocalization of Calb2 and RFP on an adjacent section.

DOI: https://doi.org/10.7554/eLife.42388.003

The following figure supplements are available for figure 2:

**Figure supplement 1.** Relating progenitor cell groups to their original positions in the cerebellum.
DOI: https://doi.org/10.7554/eLife.42388.004

**Figure supplement 2.** Examination of MidO cells along the dorsal midline of the developing cerebellum.
DOI: https://doi.org/10.7554/eLife.42388.005

until E16.5, they lacked *Lmx1a* transcripts (*Figure 2I*, *Figure 2—figure supplement 1N and O*). This is in agreement with the scenario that MidO cells originate from the roof plate where *Lmx1a* is expressed (*Chizhikov et al., 2006*), and carry residual Lmx1a protein. The lack of cell proliferation of MidO cells may contribute to the persistence of Lmx1a protein.

## Spatiotemporal changes in gene expression in the cerebellar VZ

In the developing cerebellum, the generation of glial cells partially overlaps with the generation of GABAergic neurons from the VZ (*Leto et al., 2016*). Gene expression and genetic fate mapping studies have shown that Bergmann glia arise from the VZ between E13 and E14 (*Sudarov et al.,*

*2011*; *Yuasa, 1996*). The molecular control of the parallel production of glia and GABAergic neurons from the cerebellar VZ around E13.5 is poorly understood. We reasoned that cells from multiple cerebella at a single stage might contain cells of different differentiation states due to developmental asynchrony. We explored using pseudotemporal analysis to order cells according to gene expression changes and thereby to reconstruct a developmental trajectory (*Shin et al., 2015*; *Trapnell et al., 2014*). We applied Monocle two algorithm (*Qiu et al., 2017b*; *Trapnell et al., 2014*) to the presumptive NPC plus the emerging GABAergic progenitors (cluster 3, 5, 6, and eight in *Figure 1A*). Monocle decomposed the cells into a two-phase trajectory, with one branch corresponding to Bergmann glia-like cells and the other corresponding to GABAergic precursors (*Figure 3A,B*, and *Figure 3—figure supplement 1A*). To examine genes with significant branch-dependent expression, we performed branch expression analysis modeling (*Qiu et al., 2017a*), and identified 1314 genes that were significantly different between these two trajectories (FDR < 1%). As expected, genes whose transcription increased along these two branches were enriched for gliogenesis and neuronal differentiation, respectively (*Figure 3—figure supplement 2A*). We have recently shown that Etv4 and Etv5 act downstream of Fgf-ERK signaling in the induction of Bergmann glia (*Heng et al., 2017*). Monocle correctly inferred an elevation in the expression of *Etv4* and *Etv5* prior to the upregulation of Bergmann glial markers such as *Hopx*, *Fabp7*, and *Ptprz1* (*Figure 3B*). In the presumptive GABAergic branch, the inferred transcriptional waves mostly correlated with known expression order of *Ptf1a/Kirrel2 > Olig2 > Lhx1/5 > Neurog1* during the commitment of GABAergic neurons (*Ju et al., 2016*; *Seto et al., 2014b*). Therefore, pseudotemporal ordering resolves the transcription landscapes in cerebellar NPCs as they progress over time. Remarkably, pseudotemporal ordering of the NPCs before the branch point accurately predicted gene expression changes in the VZ along the anteroposterior axis in E13.5 cerebella (*Figure 3—figure supplement 3A and B*). This demonstrates that pseudotemporal analysis can also resolve spatial expression changes in certain contexts.

The NPC group 3 cells, which are equivalent to cluster six in the full dataset (*Figure 1A* and *supplementary file 1*), display mixed molecular features of the VZ (eg. *Hes1* and *Vim*), RL (*Atoh1* and *Barhl1*), and roof plate (*Lmx1a* and *Wnt1*). As we could not reliably partition this cell group, we attempted to use pseudotemporal analysis to examine their developmental trajectories. We applied Monocle to the NPCs before the branch point one including C1 cells (*Figure 3A*). This resulted in a trajectory tree with two major branches (*Figure 3C* and *Figure 3—figure supplement 1B*). Branch expression analysis modeling identified 45 genes with significant branch-dependent expression (FDR < 1%, *Figure 3—figure supplement 3C*). Notably, cells before the branch expressed genes that were detected in the cerebellar VZ, whereas cells along the branches expressed genes that were present in the subpial stream and the roof plate, respectively (*Figure 3D and E*). Our data suggest that the posterior-most region of the cerebellar VZ represents a dynamic germinal zone containing bipotent progenitors for the RL and roof plate (*Figure 3—figure supplement 1C*). We named this region as the 'posterior transitory zone'.

## Molecular feature of different GABAergic neuron subtypes in the cerebellum

GABAergic neurons, including PCs and cerebellar nuclear interneurons, are born between E10.5 and E13.5 in mice (*Leto et al., 2016*). Refined clustering of the presumptive GABAergic lineage (cluster 13–19 in *Figure 1A*) revealed nine cell groups (*Figure 4A and B*). We tentatively assigned these cell groups as GABAergic precursors (1-2), premature GABAergic neurons (3), cerebellar nuclear interneurons (4), and PC subgroups (5-9). Groups 1 and 2 express *Kirrel2*, a marker of committed GABAergic precursors (*Mizuhara et al., 2010*). Notably, these two cell groups display different combinatory expression of *Ptf1a* and proneural genes *Ascl1*, *Neurog1*, and *Neurog2* (*Figure 4B*), reflecting the differential expression of these genes in the cerebellar VZ (*Figure 4C*) (*Zordan et al., 2008*). Although interneurons and PCs express some common markers, such as *Foxp2* and *Gad2*, the former specifically express *Pax2*, *Gad1*, *Pnoc*, and *Glra2*, whereas the latter express *Tle1* and *Islr2* (*Figure 4B*). The five newly identified PC subgroups are distinguished by the strong expression of *Etv1*, *Nrgn*, *En1*, *Cck*, and *Foxp1*, respectively (*Figure 4B and D*). Many previously known markers that define PC clusters before birth (*Fujita et al., 2012*; *Larouche et al., 2006*; *Millen et al., 1995*) are differentially expressed among the newly identified PC subgroups (*Figure 4D*). Notably, these PC subgroups exhibit variable levels of *Foxp2*, and only one expresses both *Foxp1* and *Foxp2*, which

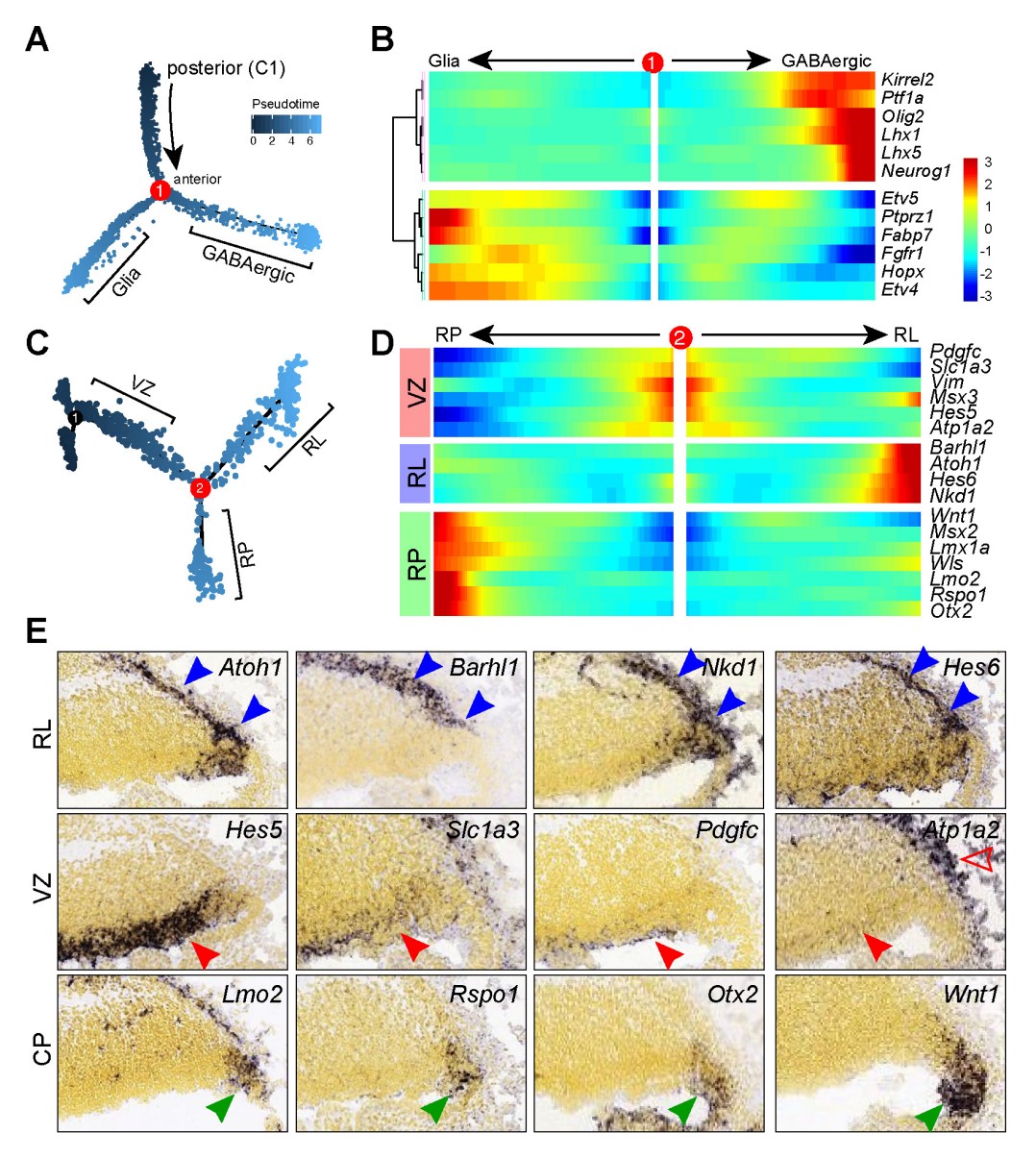

**Figure 3.** Pseudotemporal analysis reveals developmental trajectories of progenitors in the cerebellar VZ. (**A**) Monocle-inferred developmental trajectory. (**B**) Branched heatmap showing gene expression in Bergmann glial and GABAergic cells. (**C**) Monocle-inferred developmental trajectory of NPCs before the branchpoint in A. (**D**) Branched heatmap showing expression of genes that are expressed in the cerebellar ventricular zone (VZ), rhombic lip (RL) and roof plate (RP). (**E**) ISH on sagittal sections of E13.5 cerebella (from Allen Developing Mouse Brain Atlas). Arrowheads indicate transcripts in the VZ (red), RL and subpial stream (blue), and RP (green). The empty arrowhead shows expression in the meninges.

DOI: https://doi.org/10.7554/eLife.42388.006

The following figure supplements are available for figure 3:

**Figure supplement 1.** Developmental trajectories of the posterior transitory zone.
DOI: https://doi.org/10.7554/eLife.42388.007
**Figure supplement 2.** Validation of branch-dependent cascade genes.
DOI: https://doi.org/10.7554/eLife.42388.008
**Figure supplement 3.** Gene expression in the posterior end of the of cerebellar ventricular zone.
DOI: https://doi.org/10.7554/eLife.42388.009

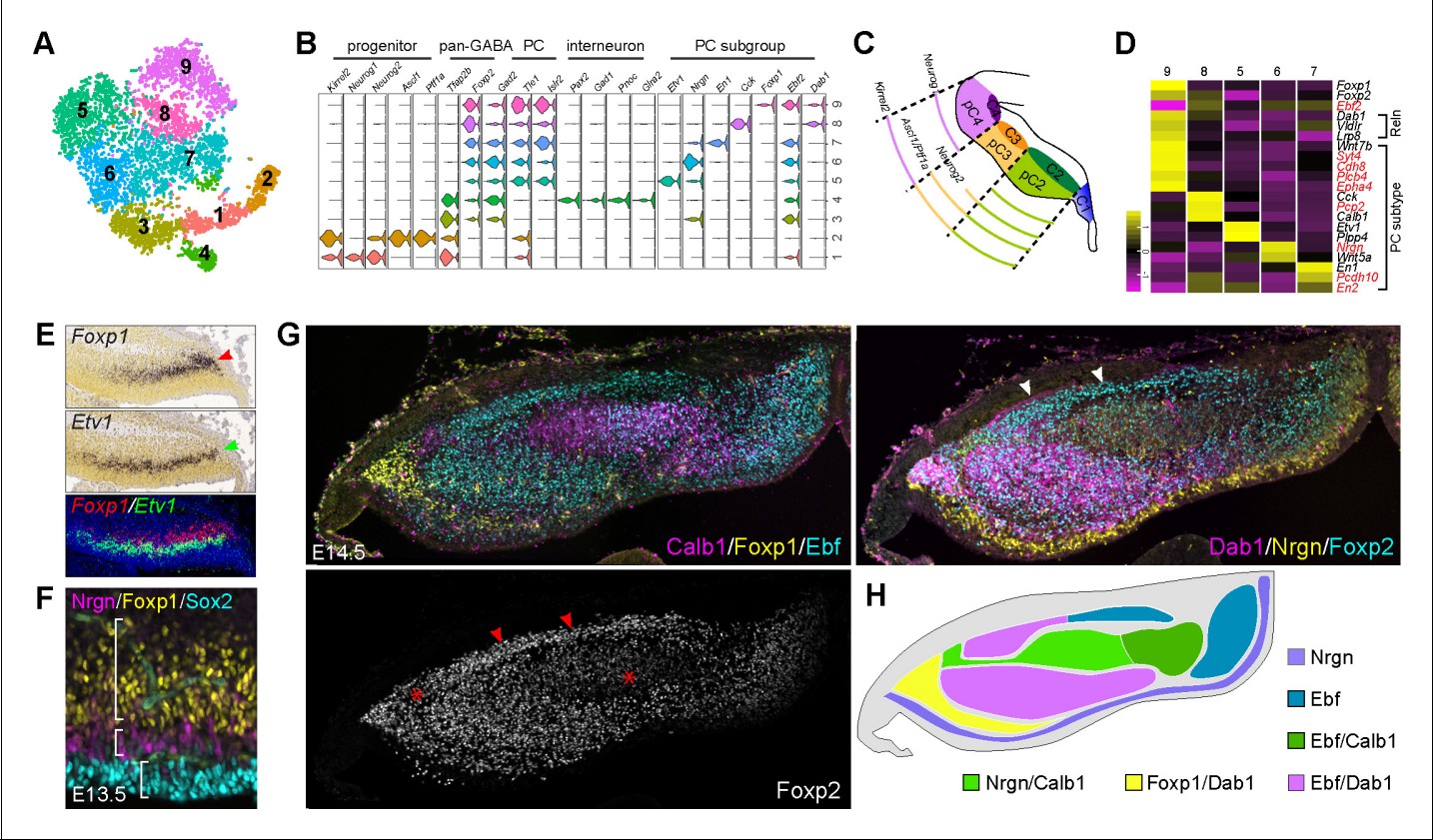

**Figure 4.** Identification of molecular features of different PC subgroups. (**A**) tSNE showing different cell groups of the cerebellar GABAergic lineage. (**B**) Violin plots showing cell-specific markers. (**C**) Schematic presentation of division of the cerebellar VZ at E12.5 (*Zordan et al., 2008*). (**D**) Heatmap showing average gene expression among PC subgroups. The known PC-subtype markers are shown in red. (**E**) ISH on coronal sections of E13.5 cerebella (from Allen Developing Mouse Brain Atlas). Red and green arrowheads indicate *Foxp1* and *Etv1* mRNAs. The overlay with pseudo-color images is shown on the bottom. (**F, G**) IHC on sagittal (**F**) and coronal (**G**) cerebellar sections of the indicated stages. The brackets in F indicate the complementary expression of Sox2, Nrgn, and Foxp1; the asterisk shows PCs with a lower level of Foxp2; the arrowhead denote the Purkinje cell plate with robust Foxp2 and Dab1 expression. (**H**) Schematic summary of Foxp2+ PC clusters based on the combination of Calb1, Dab1, Ebf, Foxp1, and Nrgn. Note that the anti-Ebf antibody may react to Ebf1-4 proteins.

DOI: https://doi.org/10.7554/eLife.42388.010

The following figure supplement is available for figure 4:

**Figure supplement 1.** Subgroups of Purkinje neurons express different levels of Foxp2 and/or different combinations of Foxp2 and Foxp1.

DOI: https://doi.org/10.7554/eLife.42388.011

specifically lacks *Ebf2* expression (*Figure 4D*). The *Foxp1*-positive (*Foxp1+*) PCs robustly express RELN receptor genes, *Vldlr*, *Lrp8*, and obligatory adaptor *Dab1* (*Figure 4D*).

We confirmed the complementary expression patterns between *Etv1* and *Foxp1* (*Figure 4E*), and between Foxp1 and Nrgn in E13.5 cerebella (*Figure 4F*). In agreement with our scRNAseq, multiple PC clusters are identified with different levels of Foxp2 expression, and distinct combinations of Calb1, Dab1, Ebf, Foxp1, Foxp2, and Nrgn expression at E14.5 (*Figure 4G*). Furthermore, IHC for Foxp1, Foxp2, and Nrgn revealed multiple PC clusters at E18.5 (*Figure 4—figure supplement 1A and B*). Foxp1 and Foxp2-double positive (Foxp1+/Foxp2+) PCs are restricted to the lateral-most region of the cerebellar hemisphere, with most in crus II of the ansiform lobule (*Figure 4—figure supplement 1C and E*). In the anterior-most region, Foxp1+/Foxp2+ PCs are present in all lobules of the cerebellar hemisphere, except for the copula pyramidis (*Figure 4—figure supplement 1D and F*). Our data suggest that multiple PC subtypes are specified shortly after PCs are born, and the acquired molecular features dictate the settlement of PC subtypes in the cerebellar cortex.

## Heterogeneity of the presumptive glutamatergic lineage

Between E10.5 and E12.5, glutamatergic neurons arise from the RL, and migrate rostrally in the subpial stream (*Fink et al., 2006*), or the RL migratory stream (*Rose et al., 2009*; *Wang et al., 2005*), to enter the nuclear transitory zone (NTZ) at the rostral end of the cerebellar anlage. From the NTZ, neurons subsequently descend deep inside the cerebellum to form fastigial, interposed, and dentate nuclei (*Elsen et al., 2012*). After E12.5, the subsequent wave of RL-derived cells become granule cell precursors, which are highly proliferative and form the external granule layer covering the surface of the developing cerebellum (*Machold and Fishell, 2005*). To examine the diversity of RL-derived neurons, we reiterated clustering of presumptive glutamatergic cell lineage (clusters 4, 10–12 in *Figure 1A*), including C1 cells (cluster 6), which express many RL-related markers (e.g. *Atoh1* and *Pax6*). Cell clustering resulted in 10 cell groups (*Figure 5A*). According to the Allen Developing Mouse Brain Atlas (*Thompson et al., 2014*), these cell groups are related to spatially distinct domains in the E13.5 cerebellum, except for two cell groups (2 and 3) that represent proliferating granule cell precursors in the S-, and M-phase of the cell cycle (*Figure 5A–C*). Cell groups 4–10 are mapped to the anterior part of the cerebellar anlage, including the NTZ. A small *Isl1*+ cell group in the anterior part of the cerebellar anlage appears contiguous with *Isl1*+ cells in the subpial region of the mesencephalon (*Figure 5C*). A subset of *Isl1*+ cells (group 4) express *Dlk1* and *Tlx3* in the anterior part of the cerebellar anlage, whereas the others (group 9) are positive for *Sncg* extending to a position ventral to the NTZ (*Figure 5C*). The NTZ can be divided into anterior and posterior compartments, which express *Lhx9* and *Neurod1*, respectively (*Figure 5C*). A discrete cell population transcribes *Atoh1* and *Th* in the anterior limit of the NTZ (*Figure 5C*). The posterior NTZ compartment can be further divided into a ventral-lateral domain positive for *Olig2*, and a dorsal-medial

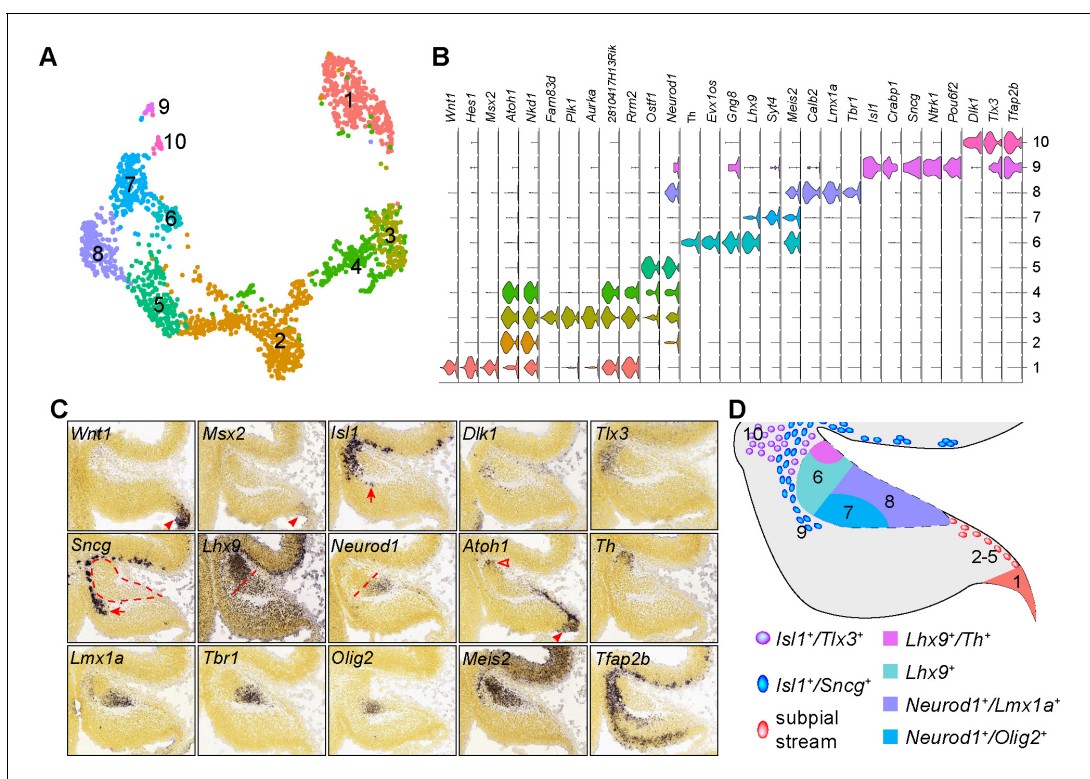

**Figure 5.** Identification of the molecular features of different cell groups that are presumably derived from the RL. (**A**) Visualization of 10 cell groups using uniform manifold approximation projection (UMAP). (**B**) Violin plots showing cell-specific markers. (**C**) ISH on sagittal section of E13.5 cerebella (from Allen Developing Mouse Brain Atlas). Arrowheads indicate mRNAs in the C1/RL; the empty arrowhead shows the second *Atoh1*+ domain at the isthmus; arrows denote *Isl1*+/*Sncg*+ + ventral to the NTZ (outlined by dashed line); the straight dashed line delineates the anterior and posterior compartments of the NTZ. (**D**) Schematic summary of the spatial distribution of the cell groups identified by scRNAseq.
DOI: https://doi.org/10.7554/eLife.42388.012

domain positive for *Lmx1a* and *Tbr1* (*Figure 5C*). Therefore, our scRNAseq has revealed highly heterogeneous cell populations in the anterior part of the cerebellar anlage at E13.5 (*Figure 5D*).

## Complex developmental origins of the cell groups in the anterior cerebellar anlage

Next, we attempted to define the origin of the cell groups identified by scRNAseq in the anterior cerebellar anlage. To this end, we performed inducible genetic fate mapping using *Gbx2creER* and *Fgf8creER* mouse lines together with a *Rosa26* Cre-reporter strain harboring tdTomato (*R26RFP*) (*Madisen et al., 2010*). *Gbx2* is broadly expressed in the VZ of rhombomere 1 encompassing the isthmus at E8.5 (*Li et al., 2002*; *Sunmonu et al., 2011*). The *Gbx2* expression domain is subsequently retracted anteriorly, and becomes absent from the RL by E10.5 (*Figure 6—figure supplement 1A*). *Fgf8* is transcribed in a transverse ring corresponding to the isthmus at E10.5 (*Figure 6—figure supplement 1A*). As expected, the descendants of *Gbx2*-expressing cells labeled at E8.5 were found throughout the cerebellar anlage (*Figure 6—figure supplement 1B*), whereas the *Gbx2* lineage labeled at E10.5 was specifically absent from the RL, subpial stream, and the posterior part of the NTZ (*Figure 6A–D* and *Figure 6—figure supplement 1C*). The *Fgf8* lineage labeled at E10.5 was restricted to the isthmus and anterior limit of the NTZ (*Figure 6E and F*). Together, these fate-mapping experiments allow identification of cells from the entire cerebellar VZ including the RL (*Gbx2*-expressing cells at E8.5), from the anterior VZ but not from the RL (*Gbx2*-expressing cells at E10.5), or from the isthmus (*Fgf8*-expressing cells at E10.5).

We found that most Isl1+ cells were negative for RFP in E13.5 *Gbx2creER/+*; *R26RFP/+* embryos that received tamoxifen at E8.5, or *Fgf8creER/+*; *R26RFP/+* embryos that received tamoxifen at E10.5 (*Figure 6A,B,E and F*). Moreover, many Isl1+ cells, particularly those close to the mid-hindbrain border, were positive for Otx2 (*Figure 6B*), whose transcription is restricted to the mesencephalon (*Millet et al., 1996*). Our data suggest that the Isl1+ cells originate from the mesencephalon to enter the cerebellum carrying residual of Otx2 protein. To determine the fate of the mesencephalon-derived Isl1+ cells, we performed inducible genetic fate mapping using an *Isl1creER* mouse line (*Laugwitz et al., 2005*). We found that the descendants of *Isl1*-expressing cells labeled at E12.5 persisted in the superior medullary velum in *Isl1creER/+*; *R26RFP/+* embryos at E16.5 (*Figure 6G and G′*). The fate-mapped cells did not contribute to the cerebellar nuclei, which were labeled by Meis2 or Lmx1a (*Figure 6H* and data not shown).

Remarkably, the descendants of *Gbx2*-expressing cells labeled at E10.5 occupied the anterior part of the NTZ, where cells strongly expressed Meis2 (*Figure 6C and D*). By contrast, the posterior part of the NTZ was mostly devoid of RFP+ cells and contained cells with weak Meis2 expression (*Figure 6D* and inset). Furthermore, RFP-labeled *Fgf8* descendants were positive for Th, confirming that the Th+ groups identified by scRNAseq represent neurons of isthmic nuclei (*Figure 6F*). Collectively, our data suggest that cells in the anterior and posterior compartments of the NTZ at E13.5 are derived from the VZ and RL, respectively.

## Expression dynamics in the RL lineage

It has been shown that *Lmx1a* and *Olig2* are differentially expressed in the medial (fastigial nuclei) and lateral (interposed and dentate nuclei) cerebellar nuclei (*Chizhikov et al., 2006*; *Ju et al., 2016*; *Seto et al., 2014a*; *Yeung et al., 2016*). Remarkably, *Lmx1a* + and *Olig2*+ cells occupy the posterior compartment of the NTZ (*Figure 5C and D*), indicating that the posterior, rather than the anterior, compartment of the NTZ contains RL-derived cerebellar nuclear neurons in accordance with our fate-mapping result (*Figure 6A–F*). To gain insight into the transcriptional regulation in RL-lineage diversification, we applied a new trajectory reconstruction algorithm, URD (*Farrell et al., 2018*), to analyze the RL-derived cells (clusters 1–5,7,8 in *Figure 5A*). This resulted in a trajectory tree that started with the posterior transitory zone and progressively diversified into granule cells, medial, and lateral cerebellar nuclear neurons, reflecting in vivo development (*Figure 7A*). Examination of *Atoh1*, *Ostf1*, *Lmx1a*, and *Lhx9*, which are known markers, showed progressive change in expression levels along each segment of the trajectory tree, as expected (*Figure 7—figure supplement 1A*). We identified 80, 36, and 47 genes that were differentially expressed along the specification trajectories forming granule cells, medial, and lateral cerebellar nuclei, respectively (*Figure 7—figure supplement 1B–D*). We also identified markers specific for granule cells, medial and lateral cerebellar

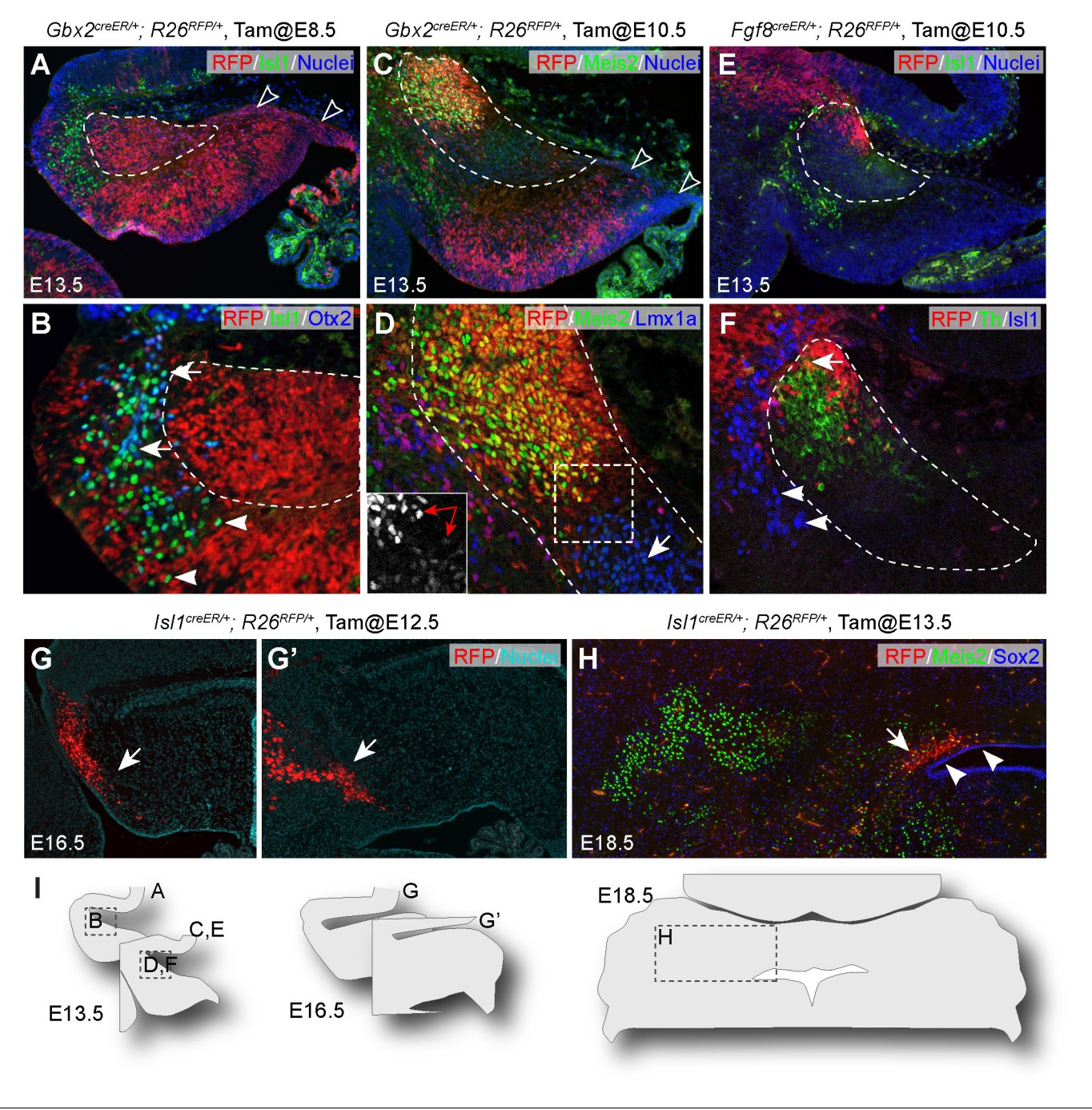

**Figure 6.** Multiple origins of cells in the anterior part of the cerebellar anlage. (A–F) IHC on sagittal sections of E13.5 brains with tdTomato red fluorescent protein (RFP) fate-mapped cells. The genotype, tamoxifen-administration stage and the antibodies used are indicated on the top and upper right corner of the image, respectively. The NTZ is outlined with dashed lines; empty arrowheads indicate the RL and subpial stream cells; arrows denote colocalization between Isl1 and Otx2 (B), between Meis2 and Lmx1a (D), or between RFP and Th (F); arrowheads indicate Isl1[+] cells anterior and ventral to the NTZ; in inset of D, the double arrow indicates cells in the anterior and posterior compartments of the NTZ express different levels of Meis2. (G and G') Serial sagittal sections of E16.5 *Isl1^{creER/+};R26^{RFP}* embryo that was given tamoxifen at E12.5. (H) IHC for Meis2 and Sox2 on a coronal section of E18.5 *Isl1^{creER/+};R26^{RFP}* embryo that received tamoxifen at E13.5. Note that RFP fate-mapped cells (arrows) are present in the superior medullary velum, but not in cerebellar nuclei, which are marked by Meis2. Arrowheads indicate the Sox2[+] cells overlaying the ventricle. (I) Illustrations indicating the relative position of images in this figure.

DOI: https://doi.org/10.7554/eLife.42388.013

The following figure supplement is available for figure 6:

*Figure 6 continued on next page*

*Figure 6 continued*

**Figure supplement 1.** The Gbx2 expression domain is excluded from the posterior end of the cerebellar ventricular zone by E10.5.

DOI: https://doi.org/10.7554/eLife.42388.014

nuclear neurons (*Figure 7—figure supplement 1E*). Confirming the validity of our analysis, we found that the transcriptional regulators that displayed significant changes along the trajectories included known regulatory genes (*Figure 7B–D*), such as *Barhl1*, *Hes6*, *Insm1*, *Kif10*, *Neurod1*, *Nhlh1*, and, *Zic1/4*, in the granule cells (*Blank et al., 2011*; *Kawauchi and Saito, 2008*; *Li et al., 2004a*; *Miyata et al., 1999*; *Zhang et al., 2018*) (*Alvarez-Rodríguez et al., 2007*; *Uittenbogaard et al., 1999*), *Pax6*, *Tbr1* and *Lmx1a* in the medial cerebellar nuclei (*Chizhikov et al., 2006*; *Fink et al.,*

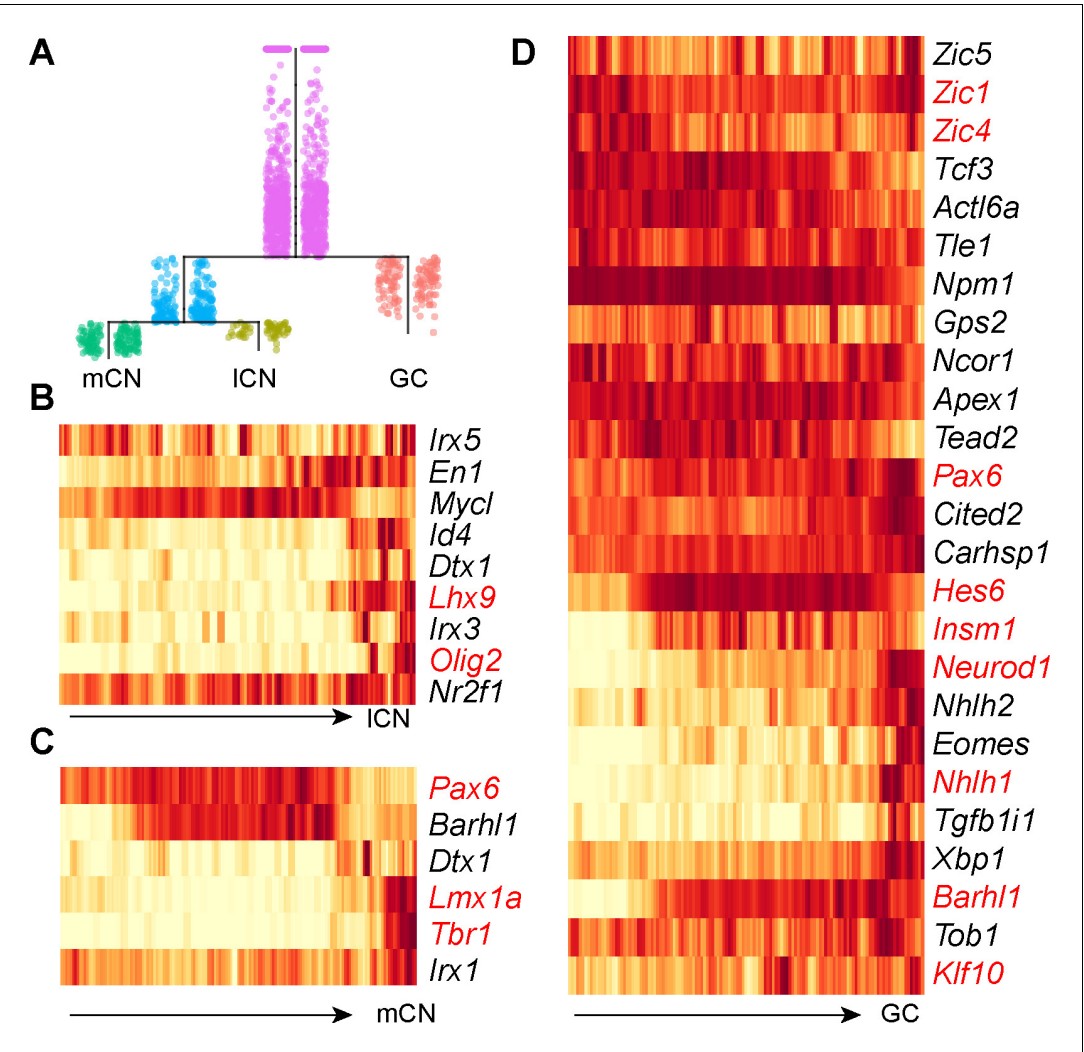

**Figure 7.** Pseudotemporal analysis reveals developmental trajectories of RL derivatives. (**A**) URD-inferred developmental trajectories. (**B–D**) Heatmap showing the dynamic expression changes of transcriptional regulators in pseudotime-ordered cells destined for lateral cerebellar nuclei (lCN), medial cerebellar nuclei (mCN), and granule neurons (GC). The transcription factors that have previously been linked to a particular trajectory are shown in red.

DOI: https://doi.org/10.7554/eLife.42388.015

The following figure supplement is available for figure 7:

**Figure supplement 1.** Distinct gene expression in the derivatives of the rhombic lip.

DOI: https://doi.org/10.7554/eLife.42388.016

*2006*; *Yeung et al., 2016*), *Olig2* and *Lhx9* in the lateral cerebellar nuclei (*Green and Wingate, 2014*; *Ju et al., 2016*; *Rose et al., 2009*; *Seto et al., 2014a*). In addition, the inferred gene cascades contained genes that were not previously associated with the development of RL derivatives. Our analyses have uncovered the transcriptional cascades, including both previously characterized and novel regulator genes, which are associated with the cell-fate specification and differentiation of granule cells and cerebellar nuclear neurons.

## Cerebellar cell types exhibit coherence expression signatures before E17.5

One major challenge in scRNAseq is to discern whether a given cell cluster represents a stable cell type or a transient cell state. While we were preparing this manuscript, a study of scRNAseq of mouse cerebella at different embryonic stages was published (*Carter et al., 2018*). We integrated ours with the published datasets (E14.5, E15.5 and E16.5) to determine whether the cell groups identified at E13.5 persisted at the later stages. To that end, we first summarized our cell clustering results and annotated all cell groups in our full dataset (*Figure 8A*). We then applied the Seurat3 algorithm (*Stuart et al., 2018*) to integrate and compare our scRNAseq data with the published one. Despite the significant technical variations among the datasets, Seurat3 successfully recovered matching cell groups among the datasets (*Figure 8B*). Remarkably, all cell groups identified in our E13.5 were detected at the later stages (*Figure 8C–D*), indicating that many of the cell groups characterized above represent canonical cell types or subtypes. As expected, granule cell precursors and glial cells are dramatically increased at the later stages, while PCs and cerebellar nuclear neurons, which are born before E14.5 become underrepresented in the E15.5 and E16.5 datasets (*Figure 8D and E*). These results suggest that cerebellar cell types, once they are born, display coherent molecular signatures until at least E16.5.

## Discussion

### Fgf17 is expressed in a signaling center along the midline of the developing cerebellum

The cerebellum comprises a seamless medial segment, called the vermis, straddling the dorsal midline. Agenesis or hypoplasia of the cerebellar vermis is a common developmental defect of the cerebellum in humans and rodents (*Basson and Wingate, 2013*; *Millen and Gleeson, 2008*) (*Aldinger and Doherty, 2016*), suggesting that the cerebellar vermian formation is intricate and susceptible to perturbations. In mice, two bilateral wing-like cerebellar anlagen meet at the dorsal midline, and the cerebellar medial regions are subsequently expanded and transformed into a homogeneous cylindrical vermis at E15.5 (*Altman and Bayer, 1997*). Remarkably, the cerebellar anlage undergoes an orthogonal rotation such that the anteroposterior axis at E9.5 becomes the mediolateral axis of the future vermis during the later stages (*Sgaier et al., 2005*). It is well documented that *Fgf8*, which is expressed in a transverse ring at the isthmus, controls the anteroposterior patterning of the cerebellar anlage and the developing midbrain (*Sato et al., 2004*). Perturbations of *Fgf8* signaling invariably result in malformation of the cerebellar vermis (*Chi et al., 2003*; *Sato et al., 2004*; *Trokovic et al., 2003*). Yet, it is mostly unknown how the anteroposterior information imparted by Fgf8 is translated to mediolateral patterning in vermian formation. In agreement with previous findings (*Alexandre, 2003*; *Cheng et al., 2012*; *Chizhikov et al., 2006*), our data suggest that roof-plate derivatives contribute to, and persist in, the midline of the developing cerebellum. Importantly, we show that these midline cells produce Fgf17, which is associated with robust cell proliferation and neurogenesis through ERK activation in the surrounding cells along the midline of the developing cerebellum (*Figure 2D–F*). We surmise that the roof-plate-derived cells act as a signaling center, the MidO. In support of this notion, mouse and human genetic studies have demonstrated that FGF17 is important for the development of the cerebellar vermis (*Lancioni et al., 2010*; *Xu et al., 2000*; *Yu et al., 2013*; *Zanni et al., 2011*). It has been shown that *Fgf8* is essential for the induction of the isthmic node, which contributes cells to the presumed MidO (*Alexandre, 2003*; *Louvi et al., 2003*). Moreover, *Fgf8* is essential for the induction of *Fgf17* (*Chi et al., 2003*; *Liu et al., 2003*). Based on these observations, we propose a new model of Fgf function in vermian formation, in which *Fgf8* controls the formation of the MidO and/or *Fgf17* expression in the

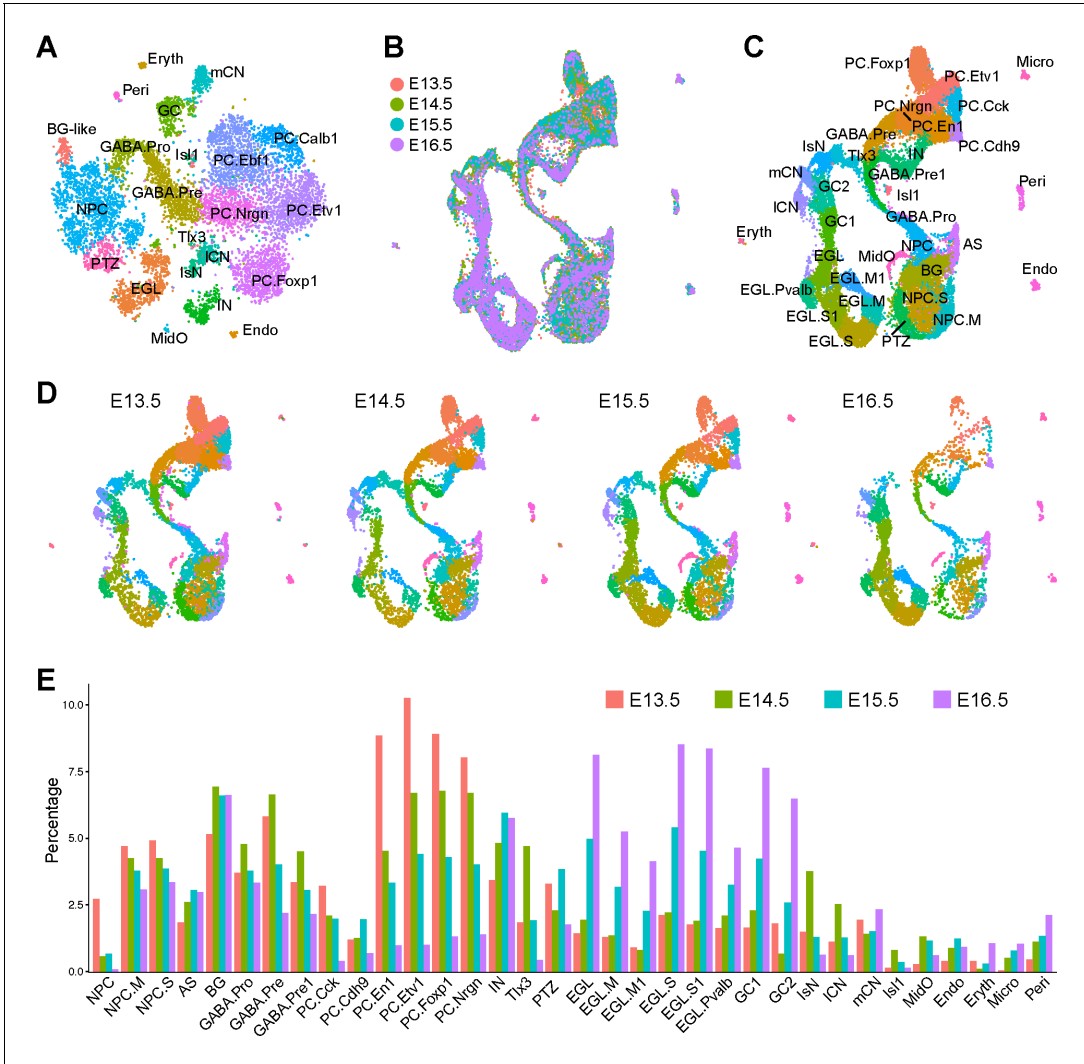

**Figure 8.** Integration of scRNAseq data from different embryonic stages. (A) Annotation of 22 cell groups recovered from the E13.5 scRNAseq dataset. (B) Integration of scRNAseq data from E13.5, E14.5, E15.5, and E16.5, with a total of 29,006 cells. The last three datasets are from published data (*Carter et al., 2018*). (C and D) UMAP visualization of cell groups in the integrated (C) and individual datasets (D). (E) Histogram showing the percentage of cell groups recovered by scRNAseq at different embryonic stages. Abbreviations: AS, astrocyte; BG, Bergmann glia; Endo, endothelium; EGL/EGL.M/EGL.S/EGL.Pvalb, external granule layer cells in different cycle phases or those with strong *Pvalb* expression; Eryth, erythrocyte; GABA.pro, GABAergic progenitor; GABA.pre/GABA.pre1, GABAergic precursors; IN, GABAergic interneurons; IsN, isthmic nuclear neurons; mCN, medial cerebellar nuclei; lCN, lateral cerebellar nuclei; NPC/NPC.M/NPC.S, neural progenitor cell and NPC at the mitotic or synthesis phase; GC1/GC2, granule cells; MidO, midline organizer cells; Micro, microglia; PC.Cck/PC.Cdh9/PC.Etv1/PC.En1/PC.Foxp1/PC.Nrgn, Purkinje cell subtype that are defined by robust Cck, Cdh9, Etv1, Foxp1 and Nrgn, respectively; Peri, pericyte; PTZ, posterior transitory zone; Isl1/Tlx3, Isl1$^+$ and Tlx3$^+$ cells found in the anterior cerebellum.

DOI: https://doi.org/10.7554/eLife.42388.017

MidO, which in turn regulates the mediolateral patterning in the formation of the cerebellar vermis. Although abnormal development of the roof plate has been linked to malformation of the vermis (*Basson and Wingate, 2013*), our findings underscore a novel mechanism that potentially contributes to congenital hypoplasia of the cerebellum associated with Joubert syndrome, Dandy-Walker malformation, and pontocerebellar hypoplasia.

## The posterior transitory zone is a dynamic germinal zone with bipotent progenitors that contribute to the rhombic lip and the roof plate

Although the cell fate of the RL lineage has been well characterized, the exact definition of the RL is less clear. Early histological studies define the RL as the 'germinal trigone' – a term refers to the fact that this germinal matrix has three prongs, the cerebellar neuroepithelium, the external granular layer, and the choroid plexus, which is the derivative of the roof plate (*Altman and Bayer, 1997*). Others define the entire *Atoh1*-expression domain as the RL (*Machold and Fishell, 2005*). The C1 domain was originally defined as cells that arise adjacent to the roof plate and expresses *Atoh1* (*Chizhikov et al., 2006*). C1 cells do not express roof-plate markers, such as *Lmx1a*, *Wnt1,* and *Gdf7* (*Chizhikov et al., 2006*). Surprisingly, genetic fate mapping studies reveal that *Lmx1a*-, *Wnt1*-, and *Gdf7*-lineage cells contribute to both GABAergic and glutamatergic neurons in the cerebellum, rather than being confined to the roof plate and its choroid plexus derivative (*Cheng et al., 2012*; *Chizhikov et al., 2010*; *Hagan and Zervas, 2012*). This suggests that the cell-fate specification among the cerebellar VZ, RL and RP is not absolute. In the current study, our scRNAseq analyses have identified an inseparable cell cluster that exhibit mixed features of VZ, RL, and RP cells. The feature genes of this cluster are expressed at the three-way intersection between the VZ, RL and RP (*Figure 3D,E*, and *Figure 3—figure supplement 1C*). Pseudotemporal analysis suggests that this cell group, together with NPCs of the rest of the VZ, form two trajectories destined for cerebellar glutamatergic neurons and the choroid plexus, respectively. These results suggest that the posterior-end of the cerebellar VZ is a dynamic germinal zone containing bipotent progenitors for cerebellar glutamatergic neurons and the choroid plexus. We name this germinal zone as the 'posterior transitory zone'. Importantly, we show that the posterior transitory zone expresses numerous genes that are in the Wnt/ß-catenin signaling pathway, including Wnt ligands, Wnt1, Wnt3a, and Wnt9a. This suggests that the Wnt/ß-catenin signaling plays an important role in the recruitment, maintenance, and/or progression of progenitor cells through this transitory zone.

## Identifying the molecular features of subtypes of PCs shortly after they are born

PCs are the cornerstone of the cerebellar circuitry – PCs orchestrate circuit assembly by regulating the number of other cell types (*Dahmane and Ruiz i Altaba, 1999*; *Fleming et al., 2013*; *Wallace, 1999*; *Wechsler-Reya and Scott, 1999*), layering of the cerebellar cortex (*Zhao et al., 2007*), and topographic connectivity (*Matsushita et al., 1991*; *Sillitoe et al., 2009*; *Sugihara and Shinoda, 2004*; *Yamada et al., 2014*). Past studies have indicated that PCs are heterogeneous in both embryonic and adult cerebella (*Dastjerdi et al., 2012*; *Sillitoe and Joyner, 2007*). Different PC subtypes display distinct susceptibilities to environmental insults or genetic mutations (*Sarna and Hawkes, 2003*). Our scRNAseq study provides the first unbiased and systematic characterization of the molecular features of newly born PCs. We identify at least five molecularly distinct PC subgroups, in agreement with the notion that PC subtypes are specified at, or immediately after their terminal differentiation (*Hashimoto and Mikoshiba, 2003*; *Karam et al., 2000*; *Larouche et al., 2006*). Importantly, the molecular features identified by scRNAseq persist among PC clusters at the later stages, suggesting that the scRNAseq-identified PC groups represent canonical subtypes, rather than transient developmental states. Notably, the identified PC subgroups display distinctive levels and combinations of *Foxp1* and *Foxp2*, which encode members of Forkhead box P subfamily of transcription factors. In vitro evidence suggests that Foxp proteins regulate distinct transcriptional targets by forming homodimers or heterodimers (*Li et al., 2004b*; *Mendoza and Scharff, 2017*; *Sin et al., 2015*). Furthermore, Foxp2 regulates PC differentiation in a dosage-dependent manner (*Fujita et al., 2008*). *Foxp1*[+]/*Foxp2*[+] PCs strongly express RELN receptors and their obligate adaptor (*Figure 4D*). This suggests that the *Foxp1*[+]/*Foxp2*[+] PC subgroup is particularly sensitive to RELN signaling, which controls PC migration (*D'Arcangelo et al., 1995*; *Howell et al., 1997*; *Sheldon et al., 1997*). Moreover, *Foxp1*[+]/*Foxp2*[+] PCs specifically lack expression of *Ebf2* (*Figure 2B*), which is essential for the Zebrin II-negative identity of PCs (*Chung et al., 2008*; *Croci et al., 2006*). Collectively, these observations suggest that the level and/or combination of Foxp1 and Foxp2 regulate the identity and settlement of PC subtypes.

## Defining the complex molecular feature and developmental origin of cells in the anterior part of the cerebellar anlage

Early histological studies suggest that postmitotic neurons delaminated from the VZ contribute to the NTZ and eventually form cerebellar nuclei (*Altman and Bayer, 1997*). Later studies show that the *Atoh1* lineage contributes to the NTZ (*Machold and Fishell, 2005*; *Wang et al., 2005*). A sub-population of the *Atoh1* lineage express *Lhx9* in the NTZ, and may contribute to interposed and dentate nuclei (*Rose et al., 2009*). On the other hand, Morales *et al*, showed that VZ-derived cells that co-express *Lhx2/9*, *Irx3*, and *Meis2* contribute to the NTZ through radial migration (*Morales and Hatten, 2006*). It is challenging to resolve the controversy because both VZ- and RL-derived cells arrive at the NTZ within a narrow window of time during early cerebellar development. Adding to the confusion is a recent discovery of a cryptic *Atoh1* expression domain at the isthmus independent of the RL (*Green et al., 2014*). Our current study has resolved the controversy by demonstrating that cells in the anterior and posterior compartments of the NTZ are differentially derived from the VZ and RL, respectively. Cells in the anterior compartment of the NTZ are mostly derived from the VZ and they robustly express Meis2 (*Figure 6C and D*). In the absence of *Ptpn11*, which encodes a protein tyrosine phosphatase that is essential for Fgf-ERK signaling, the anterior compartment of the NTZ was specifically lost; both medial and lateral cerebellar nuclei were present in *Ptpn11*-deficient cerebella at the later stages (unpublished observations by Guo et al). This supports the idea that Fgf signaling is essential for *Atoh1* expression at the isthmus but dispensable in RL derivatives (*Green et al., 2014*). Collectively, our data suggest that the *Atoh1*-expressing cells at the isthmus are derived from the cerebellar VZ.

We have previously shown that the mesencephalon contributes cells to the cerebellum (*Sunmonu et al., 2011*). Here, we have characterized the molecular features and the long-term fate of these cells. A preprint article recently described a similar population of mesencephalon-originated cells (*Rashimi-Balaei et al., 2017*). Contradictory to the idea that these mesencephalon-derived cells may contribute to cerebellar nuclei (*Rashimi-Balaei et al., 2017*), we show that these cells persist in the superior medullary velum, but not in cerebellar nuclei. The functional significance of mesencephalon-derived cells remains to be determined.

## Pseudotemporal ordering reveals gene expression landscapes in the specification of cerebellar nuclear neurons

As the sole source of output from the cerebellum, cerebellar nuclei are a critical component of cerebellar circuitry. The number of cerebellar nuclei varies among different species (*Green and Wingate, 2014*). Despite the significance of cerebellar nuclei in the function and evolution of the cerebellum, our knowledge regarding the specification and differentiation of cerebellar nuclei is woefully lacking (*Elsen et al., 2012*). Moreover, how the RL-derived *Atoh1*-lineage switches its fate from nuclear neurons to granule cells around E12.5 remains poorly understood. By exploring single cell transcription data, we describe molecular specification trajectories of the RL derivatives. The scRNAseq data and developmental trajectories provide a valuable resource for the field by describing the molecular features of newly generated neurons and global gene expression cascades towards each cell type. It will enable systematic interrogation of cell fate decision in the development of cerebellar nuclei and the changing potential of RL-derived *Atoh1* lineage. Notably, our scRNAseq data contain fewer cells for the lateral cerebellar nuclei than those for the medial cerebellar nuclei. Moreover, we are unable to partition cells for the prospective interposed and dentate nuclei. These observations suggest that the development of lateral cerebellar nuclear neurons lags behind that of medial cerebellar nuclei. Additional scRNAseq after E13.5 may be necessary to resolve the molecular features between interposed and dentate nuclei.

In summary, our scRNAseq analyses result in resolution of inter- and intra-population of heterogeneity, identification of rare cell types, and reconstruction of single-cell resolved specification trajectories during early neurogenesis of the mouse cerebellum. Using detailed histological and genetic fate mapping analyses, we not only validate the accuracy of the scRNAseq data, but also reveal the previously unappreciated complexity of the developmental origins of cerebellar neurons. Our study marks an initial but important step towards the goal of ultimately reconstructing the developmental trajectories of all cerebellar cell types.

## Materials and methods

### Mouse and tissue preparation

All procedures involving animals were approved by the Animal Care Committee at the University of Connecticut Health Center (protocol #101849–0621) and were in compliance with national and state laws and policies.. All mouse strains were maintained on an outbred genetic background. Noon of the day on which a vaginal plug was detected was designated as E0.5 in staging of embryos. Generation and characterization of the $Gbx2^{creER}$ ($Gbx2^{tm1.1(cre/ERT2)Jyhl}$/J, #022135) (*Chen et al., 2009*), $Fgf8^{creER}$ (*Hoch et al., 2015*), $R26R^{RFP}$ ($B6.Cg$-$Gt(ROSA)26Sor^{tm9(CAG-tdTomato)Hze}$/J, #007909) (*Madisen et al., 2010*), $Isl1^{creER}$ ($Isl1^{tm1(cre/Esr1*)Krc}$/SevJ, #49566) (*Laugwitz et al., 2005*), BAT-gal ($B6.Cg$-$Tg(BAT$-$lacZ)3Picc$/J, #005317) (*Maretto et al., 2003*) alleles have been described. Primer sequences for PCR genotyping and protocols are described on the JAX Mice website.

Embryonic mouse brains were dissected in ice-cold phosphate buffered saline and fixed in 4% paraformaldehyde for 40 min or overnight. Brains were cryoprotected, frozen in OCT freezing medium (Sakura Finetek), and sectioned with cryostat microtome (Leica).

### Generation of cerebellar single-cell suspensions

Brains of mouse embryos were dissected in ice-cold phosphate buffered saline. Tissues from seven animals were pooled. The dissected tissues were cut into pieces smaller than 1 mm in dimension and transferred to ice-cold MACS Tissue Storage Solution (Miltenyi Biotec, Somerville, MA). For cell dissociation, the storage solution was exchanged with RPMI 1640 Medium (Thermo-Fisher); tissue was pelleted and digested with 500 μl of pre-warmed Accumax (Innovative Cell Technologies) in a 1.5 ml tube at 37 °C for 5–10 min. At the end of digestion, the tissue pieces were dissociated by gentle trituration with a wide-bore pipet tip. The cell suspension was added to a 100 μm cell strainer (Corning, Corning, NY), and it was collected and transferred to 1.5 ml of ice-cold Resuspension Buffer (Lebovitz L15 medium with 2% FBS, 25 mM HEPES, 2 mM EDTA). The cell clumps that were unable to pass through the filter were placed into a new 500 μl pre-warmed Accumax solution, and the digestion and filtering process was repeated twice to maximize the yield of single cells. Following the dissociation, cells were stained with Trypan blue, counted and visualized with Countess II Automatic Cell Counter (ThermoFisher). The single cell suspension with over 77% viability was used to generate single cell cDNA libraries.

### Sequencing library construction and high-throughput sequencing

After single cell suspension, the 10X Genomics Chromium Single Cell Kit v1 (PN-120233) was used to create cDNA libraries. Samples were then sequenced on Illumina NextSeq 500. The raw reads were processed to molecule counts using the Cell Ranger pipeline with default settings (*Zheng et al., 2017*).

### Cell clustering and classification

The raw UMI counts from Cell Ranger was processed by the Seurat R package (version 1.4) (*Butler et al., 2018*; *Macosko et al., 2015*). Genes that were detected in less than three cells were removed. Cells in which over 5% of the UMIs were mapped to the mitochondrial genes were discarded, and cells that contained less than 200 or over 4800 genes were considered outliers and discarded. Library-size normalization was performed on the UMI-collapsed gene expression for each barcode by scaling the total number of transcripts per cell to 10,000. The data were then $\log_2$ transformed. In total, 9,306 cells and 15,823 genes (an average of 1200 detected genes/cell) were used in the cell type determination.

Linear regression was used to regress out the total number of UMIs and the fraction of mitochondrial transcript content per cell. The variable genes were identified using Seurat's *MeanVarPlot* function using the following parameters: x.low.cutoff = 0.0; y.cutoff = 0.8, resulting in 1945 variable genes. These variable genes were used in the principal component analysis. The first 23 principal components were used for cell clustering with a resolution at 0.6, and low dimensional reduction to visualize cell clusters. Specific genes for each cluster were identified using the Seurat's *FindAllMarkers* function. To refine clustering, the *SubsetData* function was used to create a new Seurat object containing only a subset of the original cells and cell clustering was reiterated.

## Integration of multiple scRNAseq datasets

We obtained the mouse embryonic scRNAseq datasets (*Carter et al., 2018*) from the European Nucleotide Archive PRJEB23051. The downloaded BAM files were converted to FASTQ files using the bamtofastq tool (version 1.1.2, 10X Genomics), and the resultant FASTQ files were processed to generate molecule counts using the Cell Ranger pipeline. The raw UMI counts was processed by the Seurat (version 3.0) (*Stuart et al., 2018*). Cells in which over 5% of the UMIs were mapped to the mitochondrial genes were discarded, and cells that contained less than 700 or over 4000 genes were considered outliers and discarded. We found that PCs and cerebellar nuclear neurons were severely underrepresented in datasets after E16.5. Therefore, we only included E14.5, E15.5, and E16.5 in the current study. Using Seurat version 3.0.0, we merged the E13.5 (9,306 cells), E14.5 (7,194 cells), E15.5 (7,335 cells), and E16.5 (5,171 cells) datasets. We used the *FindIntegrationAnchors* function to 'anchor' datasets together with default parameters (dimensionality = 1:30). The anchors were then passed through the *IntegrateData* function to generate a Seurat object with integrated (or batch-corrected) expression matrix. Standard cell clustering and visualization were performed.

## Pseudotemporal ordering

Monocle 2 (version 2.4.0) was used to infer the pseudotemporal ordering of NPCs (clusters 3, 5, 6, and eight in *Figure 1A*). We assumed that the raw UMI counts were distributed according to negative binomial, and we estimated size factors and dispersion using the default functions. Dimensionality reduction was carried out using the default DDRTree method. The Monocle differentialGeneTest or BEAM functions were used to identify genes differentially expressed along pseudotime or particular branches. Branched heatmaps were constructed using genes with q < 0.05 from the BEAM. The ward.D2 clustering method was applied on the correlation matrix for the transformed data between all the genes. To obtain the enriched biological process gene ontology term, we performed the hypergeometric test on the corresponding Gene Matrix Transposed file format (GMT) file for each cluster of genes based on the piano R package (*Väremo et al., 2013*). We selected transcription factors based on AnimalTFDB (*Zhang et al., 2015*).

The URD R package (version 1.0) was used to reconstruct the trajectories of presumptive RL derivatives (clusters 1–5,7,8 in *Figure 5A*). As described previously (*Farrell et al., 2018*), we worked backward from the tip cells (cluster 5,7,8) along the population's trajectory with cluster one as root cells. Genes were considered differentially expressed if they were expressed in at least 10% (frac.must.express = 0.1) of cells in the trajectory segment under consideration, their mean expression was upregulated 1.5x (log.effect.size = 0.4) compared to the siblings and the gene was 1.25x (auc.factor = 1.25) better than a random classifier for the population as determined by the area under a precision-recall curve. Genes were considered part of a population's cascade if, at any given branchpoint, they were considered differential against at least 60% (must.beat.sibs = 0.6) of their siblings, and they were not differentially upregulated in a different trajectory downstream of the branchpoint (i.e. upregulated in a shared segment, but really a better marker of a different population).

## In situ hybridization and immunohistochemistry

Standard protocols were used for X-gal histochemistry, immunofluorescence, and in situ hybridization, as described previously (*Chen et al., 2009*). Detailed protocols are available on the Li Laboratory website (http://lilab.uchc.edu/protocols/index.html). Primary and secondary antibodies used in the study are listed in *Supplementary file 2*. To generate riboprobe templates, cDNA of E13.5 mouse brain was used to PCR amplify the 3' end of the target gene, ranging between 500–700 bp with the T7 RNA polymerase recognition site incorporated into the product. Standard in vitro transcription methods using T7 polymerase (Promega, Madison, WI) and digoxigenin-UTP RNA labeling mix (Roche) were used to produce antisense riboprobes.

For EdU pulse-chase labeling, EdU (0.5 mg/ml in PBS) was injected intraperitoneally into pregnant female mice at 10 µg/g body weight, and embryos were dissected 24 hr later. EdU labeling was detected with the Click-iT EdU Imaging Kit (Invitrogen, Carlsbad, CA, USA).

## Data availability

The sequencing files and raw gene count matrix have been deposited in NCBI's Gene Expression Omnibus and are accessible through accession number GSE120372. All the computer codes associated with the manuscript are available in the supporting zip document and at https://github.com/JLi-Lab/scRNAseq_Cerebellum (*Wizeman et al., 2019*; copy archived at https://github.com/elifesciences-publications/scRNAseq_Cerebellum).

## Acknowledgements

We would like to thank Drs. Paul Robson and Mohan Bolisetty for their advice and technical assistance in scRNAseq. We are grateful to Dr. Alexandra Joyner for providing *Fgf8creER* mice, Dr. Jaime Rivera for BAT-gal mice, and Dr. Bennett Novitch for providing the anti-Foxp1 antibody. The monoclonal anti-Isl1 antibody was obtained through the Developmental Studies Hybridoma Bank under the auspices of the NICHD and maintained by The University of Iowa (Iowa City, IA). E Wilion was supported by the University of Connecticut Health Research Program.

## Additional information

### Funding

| Funder | Grant reference number | Author |
|---|---|---|
| NIH Office of the Director | R01NS106844 | James YH Li |
| University of Connecticut | Health Research Program | Elliot M Wilion |

The funders had no role in study design, data collection and interpretation, or the decision to submit the work for publication.

### Author contributions

John W Wizeman, Formal analysis, Validation, Investigation, Writing—original draft, Writing—review and editing; Qiuxia Guo, Data curation, Validation, Investigation, Visualization, Writing—review and editing; Elliott M Wilion, Data curation, Validation, Visualization, Writing—review and editing; James YH Li, Conceptualization, Resources, Data curation, Software, Formal analysis, Supervision, Funding acquisition, Validation, Investigation, Visualization, Methodology, Writing—original draft, Project administration, Writing—review and editing

### Author ORCIDs

James YH Li http://orcid.org/0000-0002-9231-2698

### Ethics

Animal experimentation: All procedures involving animals were approved by the Animal Care Committee at the University of Connecticut Health Center and were in compliance with national and state laws and policies. (protocol #101849-0621

### Decision letter and Author response

Decision letter https://doi.org/10.7554/eLife.42388.025
Author response https://doi.org/10.7554/eLife.42388.026

## Additional files

### Supplementary files

• Supplementary file 1. Molecular features of the identified cell clusters. Results of differential expression analysis of identified clusters of all cells (first spreadsheet), neural progenitor cells (second spreadsheet), GABAergic neurons and the precursors (third spreadsheet), and glutamatergic neurons and their precursors (forth spreadsheet).

DOI: https://doi.org/10.7554/eLife.42388.018

• Supplementary file 2. Functional and pathway enrichment analyses of feature genes of NPC clusters. Results of functional and KEGG pathway enrichment analysis.
DOI: https://doi.org/10.7554/eLife.42388.019

• Supplementary file 3. List of antibodies, primers used in the current study. Lists of primary and secondary antibodies used in the current study.
DOI: https://doi.org/10.7554/eLife.42388.020

• Transparent reporting form
DOI: https://doi.org/10.7554/eLife.42388.021

## Data availability

Sequencing data have been deposited in GEO under accession codes GSE120372. All the computer codes associated with the manuscript are available in the supporting zip document and at https://github.com/JLiLab/scRNAseq_Cerebellum (copy archived at https://github.com/elifesciences-publications/scRNAseq_Cerebellum).

The following dataset was generated:

| Author(s) | Year | Dataset title | Dataset URL | Database and Identifier |
|---|---|---|---|---|
| James Li | 2018 | Sinle-cell RNA sequecing of E13.5 mouse cerebella | https://www.ncbi.nlm.nih.gov/geo/query/acc.cgi?acc=GSE120372 | NCBI Gene Expression Omnibus, GSE120372 |

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
