## [Decision Letter]

Thank you for submitting your article "Specification of diverse cell types during early neurogenesis of the mouse cerebellum" for consideration by *eLife*. Your article has been reviewed by two peer reviewers, and the evaluation has been overseen by a Reviewing Editor and Marianne Bronner as the Senior Editor. The reviewers have opted to remain anonymous.

The reviewers have discussed the reviews with one another and the Reviewing Editor has drafted this decision to help you prepare a revised submission.

Summary:

Wizeman et al. have performed single cell RNASeq on the developing mouse cerebellum at E13.5. This time point is one of active proliferation and cell fate determination. They use their data to model some developmental trajectories, and propose some new regulators for cerebellar ventricular zone and rhombic lip progenitor cells. A number of novel observations are made which will provide insight into, and candidates for, regulation of patterning and other key developmental events from this time period. However, as pointed out by the reviewers, there are a number of validations that are needed for their interpretations and conclusions.

Essential revisions:

1) The Mid-O data are weak and require additional expression data to support MidO distinction from the IsO. These can be provided by ISH/IHC over several time points. Suggestions for markers include: Lmx1a, Wls, Calbindin and *Fgf17* at multiple stages and after fate mapping. Some of this might be best addressed in whole mount staining to ensure the entire dorsal midline is captured instead of sections.

2) As above, additional expression data are needed to support Purkinje cell subtypes.

3) Single cell RNA data have been recently published by two groups: Gupta and Carter. It is odd that the authors have not referenced these papers. They may find additional data in these papers to back up their own conclusions. At the very least, a thorough comparison should be made with the data in these other papers.

4) There are many excellent suggestions for improvements from both reviewers. The authors should address all of the comments regarding interpretations, and provide clarity where it is asked for. These comments do not require additional experiments, other than those above.

*Reviewer #2:*

Using a combination of single cell sequencing of 9000 cells from E13.5 mouse cerebellum, the authors have defined developmental trajectories and potential new regulators for cerebellar ventricular zone and rhombic lip progenitors. The authors have thoughtfully sorted through the scRNAseq data, complemented their analyses with immunohistochemistry and fate mapping and make the following major additions to the field of cerebellar development:

1) Previously undescribed populations including the Isl1 population, distributed along the superior medullary vellum;

2) Definition of new signaling populations – the MidO;

3) Described new compartmentalization and heterogeneity of the NTZ the organizing center for Cerebellar Nuclei. This is particularly impactful data – the orchestration Cerebellar Nuclei are very poorly understood and there are important new insights;

4) Definition of early Purkinje cell heterogeneity, only previously hinted at from prior literature;

5) Further description of the RL glutamatergic lineage.

This kind of detailed analysis is needed for the cerebellum, where a limited number of cell types provide the promise of actually defining the full complement of developmental cellular and molecular regulatory pathways that comprise this important CNS region. The discussion and illustration of NTZ/anterior CB is very well done. As is the presentation and discussion of Purkinje cell heterogeneity.

In Tsne plots, cluster definitions/boundaries are not clear. How defined as distinct? Also – very confusing, referring to clusters and groups throughout manuscript.

The definition and understanding of the RL has evolved considerably since first C1 was first defined in 2006 and I strongly suggest that the authors refrain from using the C1 terminology, which is used in variety of different contexts throughout this manuscript. "RL progenitor zone" and "RL derivative populations" are more precise. RL derivatives include GCPs which are also progenitors. The published data do indeed support that the separation of fates is not absolute in agreement with the new data presented here. This is yet another reason to avoid C1 terminology. Please rewrite this section of the Discussion and elsewhere in the manuscript where appropriate. Note too that others have already shown extensive *Wnt1* expression in this zone (for example, PMID: 21857982)

Groups 3/4 of Figure 2 needs clarification: Expression of Wls and *Fgf17* suggests that this is actually a very mixed population of progenitors from the RL and anterior Isthmic region, assuming Wls is only on the RL as published by the Goldowitz group. Please provide immuno data to clarify.

Surprisingly, a highly relevant paper (Lancaster et al., 2011) was overlooked in the discussion/definition of MidO cells. This prior study implicated midline Wnt+/Batgal+ cells in vermis fusion at E135. This work must be cited/discussed.

The second last sentence of the Abstract over-emphasizes the MidO result and seems only to be included because of potential disease relevance – this is a stretch, not well explored in this work and should be de-emphasized.

*Reviewer #3:*

This paper describes an in depth computational analysis of scRNAseq of ~10,000 cells from the cerebellum at one embryonic stage, E13.5, with limited confirmation of the conclusions from RNA in situ data (Allen Institute or their own) and fate mapping. A number of interesting conclusions are made, however, they are not supported by experimental or other data. In general, the results will be of great interest to specialists in the cerebellum development field. Ideally, scRNA-seq would be done at multiple stages or at least the validation should be performed at multiple stages in mice that include fate mapping of the proposed populations. This is crucial to observe temporal regulation of some of the events described. In particular, in the section where the authors describe new signaling centers, it would be important to determine when they are established and function and are resolved.

The authors state "Refined clustering of presumptive NPCs (cluster 3, 5, 6, 8 in Figure 1A) Group 1 cells are presumably radial glia, as they express *Hes1* and *Hes5*. Group 2 cells likely represent Bergmann glial precursors, as they express many Bergmann-glia-specific genes that we recently described." Proof of this should be shown.

Similarly, the authors state "groups 3 and 4 encode molecules that are significantly enriched in the Wnt signaling and group 4 cells express *Fgf17*… Therefore, cell groups 3 and 4 may represent two signaling centers in the cerebellar anlage." All groups express some signaling molecules, therefore this is an overstatement. While the hypothesis is interesting, no evidence is shown that the small number of cells in each group act as signaling centers.

"that the descendants of *Wnt1*-expressing cells labeled at E8.5 contributed to the midline and express Lmx1a and *Calb2* at E14.5 (Figure 2I and data not shown)." Data not shown is a very critical piece of evidence. Please include. In addition it would be good to see *Calb2* and *Fgf17* expression at the midline after fate mapping.

"In addition to Wnt/ß-catenin activity, the MidO patterns the cerebellar midline via FGF-ERK signaling." It seems this is not a new organizer, but the isthmus organizer known to express *Wnt1* and *Fgf8/17/18*. The perhaps new observation is that it persists later than has been analyzed in any detail. Also, midline is not a very accurate name, since midline can mean all the cells in the cerebellum in the most medial sagittal sections, or the floorplate of the brain is considered midline. Instead the authors mean the dorsal midline in very specific coronal sections that include the cerebellum. Sagittal sections should be included to determine if it is the same as the isthmic organizer.

"This is in agreement with the scenario that MidO cells originate from the roof plate where Lmx1a is expressed, and carry residual Lmx1a proteins." How many days are the authors suggesting LMX1a protein persists? When is the transcript expressed in the MidO?

Where do the MidO cells fall in the pseudotime trajectories?

Since the pseudotime analysis, despite being performed on a single time point, identified clear populations it would be useful to project these different populations/branches identified at Figure 3A and C on the original clustering analysis performed. This will help with an understanding of the distinct populations obtained in the initial analysis.

"In the presumptive GABAergic branch, transitions from Ptf1a/Kirrel2 to Lhx1, Olig2, Lhx5 and Neurog1" Other than Ptf1a/Kirrel2, this set of genes does not seem like key genes in the GABA lineage. Why do the authors think they play a role? In situ hybridization of cells in the lineage over time should be shown, for example using Ptf1a-Cre fate mapping.

"Our data suggest that the C1 domain represents a dynamic germinal zone that arises from the VZ and contains bipotent progenitors for the anterior RL and roof plate." What do the authors mean by the anterior RL? What does this give rise to? If they mean the cells give rise to the anterior EGL, why are they expressed in the ventral half of the RL at E14.5 (Figure 3—figure supplement 2), after the anterior EGL cells have left the RL? Additional fate mapping would be needed to prove this conclusion.

"We identified 80, 36, and 47 genes that were differentially expressed along the specification trajectories forming granule cells, medial, and lateral cerebellar nuclei, respectively (Figure 7— – —figure supplement 1B-D)." Can this conclusion be confirmed with expression data at different time points?

"we show that roof-plate derivatives contribute to, and persist in, the midline of the developing cerebellum." It is not clear to me how the authors demonstrate this. Is the roof plate only in the midline? What exactly is it? In the images, the arrows point to cells near the RL, but then below they say they are at the anterior end of the cerebellum.

"Importantly, we show that these midline cells produce *Fgf17*". Are the authors suggesting roofplate cells become the midline that expresses *Fgf17*? Why not propose that *Fgf17*, which is expressed in the isthmus, stays on dorsally and becomes what they refer to as midO?

"in turn promotes cell proliferation and neurogenesis through ERK activation in the surrounding cells along the midline". The authors show no proof of this, so it should be proven or said as a suggestion.

Figure 4D. The data presented does not support the conclusion that only one of the subtypes have FoxP1/2 expression, since the IF suggests that most of the PCs express them. Please elaborate more on this. Also more validation of the other markers used to identify the 5 putative PC types is required. With the present data, the presence of these subtypes is not supported.

Figure 7. Do any of the transcription factors identified in the different lineages have common upstream regulatory signals?

---

## [Author Response]

Essential revisions:1) The Mid-O data are weak and require additional expression data to support MidO distinction from the IsO. These can be provided by ISH/IHC over several time points. Suggestions for markers include: Lmx1a, Wls, Calbindin and Fgf17 at multiple stages and after fate mapping. Some of this might be best addressed in whole mount staining to ensure the entire dorsal midline is captured instead of sections.

a) We have added two new supplementary figures to strengthen the MidO result. First, we provide additional data to show how we relate different neural progenitor cell groups, including the MidO cells, to their endogenous position in the cerebellar anlage (Figure 2—figure supplement 1). Second, we provide ISH for *Fgf17* on a serial of coronal sections of E13.5, and sagittal sections of E16.5 cerebella (Figure 2—figure supplement 2A and B). These data demonstrate that the *Fgf17* expression domains in the isthmus and MidO are different.

b) We provide images of E10.5 *Wnt1-creER; R26^RFP/+^* embryo that was administrated with tamoxifen at E8.5 in wholemount and flatmount preparations (Figure 2—figure supplement 2C and D). This shows the contribution of the *Wnt1* lineage to the MidO. As requested, we now show the expression of *Calb2* in the *Wnt1*-lineage cells along the midline (inset in Figure 2I).

c) We now incorporate and compare the published scRNAseq from Carter et al. with our E13.5 data. In a new figure, we demonstrate that persistent of MidO cells at the later stages (Figure 8).

2) As above, additional expression data are needed to support Purkinje cell subtypes.

a) We have added addition expression data to demonstrate the different Purkinje cell subtypes at E14.5 (Figure 4G).

b) By incorporating the scRNAseq data from Carter et al., we show that the different Purkinje cell subtypes identified in our E13.5 dataset are present at E14.5, E15.5 and E16.5. The new analysis demonstrates that the PC groups identified through our E13.5 scRNAseq represent canonical PC subtypes, rather than transient cell states during development.

3) Single cell RNA data have been recently published by two groups: Gupta and Carter. It is odd that the authors have not referenced these papers. They may find additional data in these papers to back up their own conclusions. At the very least, a thorough comparison should be made with the data in these other papers.

We thank the reviewers’ excellent suggestion. We have incorporated Carter et al. data to support our conclusions. We found that Purkinje cells and cerebellar nuclear neurons are severely underrepresented in available scRNAseq datasets of neonatal and postnatal stages. Therefore, we limited our analyses between E13.5 and E16.5.

4) There are many excellent suggestions for improvements from both reviewers. The authors should address all of the comments regarding interpretations, and provide clarity where it is asked for. These comments do not require additional experiments, other than those above.

We have made changes as suggested. Our point-by-point responses are listed below.

Reviewer #2:[…] In Tsne plots, cluster definitions/boundaries are not clear. How defined as distinct? Also – very confusing, referring to clusters and groups throughout manuscript.

a) We are unaware of any gold standard in cell clustering, or quantitative evaluation of the separation of cell clusters. Using repeating subsampling, we have demonstrated the robustness of cell clustering.

b) In the revised manuscripts, we have clarified the reference of cell clusters and cell groups.

a.

c) We add a new figure to summarize the annotation of all cell groups (Figure 8A).

The definition and understanding of the RL has evolved considerably since first C1 was first defined in 2006 and I strongly suggest that the authors refrain from using the C1 terminology, which is used in variety of different contexts throughout this manuscript. "RL progenitor zone" and "RL derivative populations" are more precise. RL derivatives include GCPs which are also progenitors. The published data do indeed support that the separation of fates is not absolute in agreement with the new data presented here. This is yet another reason to avoid C1 terminology. Please rewrite this section of the Discussion and elsewhere in the manuscript where appropriate. Note too that others have already shown extensive Wnt1 expression in this zone (for example, PMID: 21857982)

a) To the best of our knowledge, the initial characterization of C1 matches well with the newly identified cell group at the three-way intersection between the VZ, RL, and RP. Therefore, we continue to use this term in the initial description. Following the pseudotime analysis, we redefine this domain as “posterior transitory zone” to reflect the bipotency of the progenitor cells in this region to form both rhombic lip and roof plate.

b) We have cited papers by Selvaduri et al., 2011 and Lancaster et al., 2011.

Groups 3/4 of Figure 2 needs clarification: Expression of Wls and Fgf17 suggests that this is actually a very mixed population of progenitors from the RL and anterior Isthmic region, assuming Wls is only on the RL as published by the Goldowitz group. Please provide immuno data to clarify. Surprisingly, a highly relevant paper (Lancaster et al., 2011) was overlooked in the discussion/definition of MidO cells. This prior study implicated midline Wnt+/Batgal+ cells in vermis fusion at E13.5. This work must be cited/discussed.

a) We have provided IHC data to show that Wls is expressed in the RL progenitor domain and MidO (Figure 2—figure supplement 1M-M2).

b) The Lancaster et al. paper is now cited.

The second last sentence of the Abstract over-emphasizes the MidO result and seems only to be included because of potential disease relevance – this is a stretch, not well explored in this work and should be de-emphasized.We agree that the organizing function of MidO is only suggestive. We have removed this sentence from the Abstract.Reviewer #3:[…] The authors state "Refined clustering of presumptive NPCs (cluster 3, 5, 6, 8 in Figure 1A) Group 1 cells are presumably radial glia, as they express Hes1 and Hes5. Group 2 cells likely represent Bergmann glial precursors, as they express many Bergmann-glia-specific genes that we recently described." Proof of this should be shown.

We provide new data to relate NPC groups to their endogenous positions in the cerebellar anlage (Figure 2C and Figure 2—figure supplement 1).

Similarly, the authors state "groups 3 and 4 encode molecules that are significantly enriched in the Wnt signaling and group 4 cells express Fgf17… Therefore, cell groups 3 and 4 may represent two signaling centers in the cerebellar anlage." All groups express some signaling molecules, therefore this is an overstatement. While the hypothesis is interesting, no evidence is shown that the small number of cells in each group act as signaling centers.We agree that this statement seems out of place. The sentence is removed. On the other hand, it is worth noting that, although it is true that most cell groups express some signaling molecules, only groups 3 and 4 show statistical significance in the enrichment of genes in the Wnt signaling pathway. The enrichment test results are now shown in Supplementary file 2. Furthermore, we and others have demonstrated the Wnt/ß-catenin activity at the cerebellar midline region and the so-called “posterior transitory zone”."that the descendants of Wnt1-expressing cells labeled at E8.5 contributed to the midline and express Lmx1a and Calb2 at E14.5 (Figure 2I and data not shown)." Data not shown is a very critical piece of evidence. Please include. In addition it would be good to see Calb2 and Fgf17 expression at the midline after fate mapping.*Calb2* staining is provided (inset in Figure 2I)."In addition to Wnt/ß-catenin activity, the MidO patterns the cerebellar midline via FGF-ERK signaling." It seems this is not a new organizer, but the isthmus organizer known to express Wnt1 and Fgf8/17/18. The perhaps new observation is that it persists later than has been analyzed in any detail. Also, midline is not a very accurate name, since midline can mean all the cells in the cerebellum in the most medial sagittal sections, or the floorplate of the brain is considered midline. Instead the authors mean the dorsal midline in very specific coronal sections that include the cerebellum. Sagittal sections should be included to determine if it is the same as the isthmic organizer.

a) We agree that “midline” could be an ambiguous term when referring the neural tube. However, in the context of the cerebellar vermian formation, we believe that the midline is quite specific. Therefore, we decide to keep the name.

b) In a new supplementary figure, we show *Fgf17* ISH on serial coronal sections of the E13.5 cerebellum, and on two adjacent sagittal sections of the E16.5 cerebellum (Figure 2—figure supplement 1A and B). Our data demonstrate that the *Fgf17* expression domain in the MidO differs from that in the isthmus.

"This is in agreement with the scenario that MidO cells originate from the roof plate where Lmx1a is expressed, and carry residual Lmx1a proteins." How many days are the authors suggesting LMX1a protein persists? When is the transcript expressed in the MidO?

In the revised manuscript, we have clarified that neither ISH (Figure 2—figure supplement 2N) and scRNAseq show the presence of *Lmx1a* transcripts in the MidO. On the other hand, Lmx1a immunoreactivity is present in the MidO at E13.5 (Figure 2D), and the signal persists at least until E16.5 (Figure 2I). We suggest that the lack of mitotic activity of MidO cells may contribute to the prolonged presence of Lmx1a protein.

Where do the MidO cells fall in the pseudotime trajectories?The MidO cells were not included in the pseudotime analysis.Since the pseudotime analysis, despite being performed on a single time point, identified clear populations it would be useful to project these different populations/branches identified at Figure 3A and C on the original clustering analysis performed. This will help with an understanding of the distinct populations obtained in the initial analysis.

We thank the reviewer for the excellent suggestion. We have added a supplementary figure to show the Seurat clusters in the Monocle-inferred trajectories (Figure 3—figure supplement 1).

"In the presumptive GABAergic branch, transitions from Ptf1a/Kirrel2 to Lhx1, Olig2, Lhx5 and Neurog1" Other than Ptf1a/Kirrel2, this set of genes does not seem like key genes in the GABA lineage. Why do the authors think they play a role? In situ hybridization of cells in the lineage over time should be shown, for example using Ptf1a-Cre fate mapping.

Past studies have demonstrated the expression and/or functional role of *Ptf1a* (Hoshino et al., 2005), *Kirrel2* (Mizuhara et al., 2010), *Lhx1/5* (Zhao et al., 2007), *Olig2* (Ju et al., 2016), and *Neurog1* (Zordan et al., 2008) in the development of cerebellar GABAergic neuron. Others have characterized the temporal expression order of these genes in cerebellar GABAergic lineage (Ju et al., 2016; Seto et al., 2014). Importantly, the predicted gene expression changes based on Monocle analysis are mostly in agreement with the published data. Therefore, we don’t believe that it is necessary to repeat the expression analysis.

"Our data suggest that the C1 domain represents a dynamic germinal zone that arises from the VZ and contains bipotent progenitors for the anterior RL and roof plate." What do the authors mean by the anterior RL? What does this give rise to? If they mean the cells give rise to the anterior EGL, why are they expressed in the ventral half of the RL at E14.5 (Figure 3—figure supplement 2), after the anterior EGL cells have left the RL? Additional fate mapping would be needed to prove this conclusion.

We apologize for the confusion. We referred the upper RL as anterior RL, contrasting the lower/posterior RL that gives rise to brainstem lineages. As the current study only focuses on the cerebellar rhombic lip, we decide to use RL in referring the upper/anterior RL throughout the text.

"We identified 80, 36, and 47 genes that were differentially expressed along the specification trajectories forming granule cells, medial, and lateral cerebellar nuclei, respectively (Figure 7—figure supplement 1B-D)." Can this conclusion be confirmed with expression data at different time points?

As we described in the text, some of these genes have been linked to the respective cell types. In the revised manuscript, we have added additional scRNAseq data from the later stages to support our conclusion. We believe these data are sufficient to support our conclusion.

"we show that roof-plate derivatives contribute to, and persist in, the midline of the developing cerebellum." It is not clear to me how the authors demonstrate this. Is the roof plate only in the midline? What exactly is it? In the images, the arrows point to cells near the RL, but then below they say they are at the anterior end of the cerebellum.

We changed the sentence as “our data suggest that roof-plate derivatives contribute to…”. The cerebellar epithelium interfaces with the roof plate at the dorsal midline and throughout its posterior limit.

"Importantly, we show that these midline cells produce Fgf17". Are the authors suggesting roofplate cells become the midline that expresses Fgf17? Why not propose that Fgf17, which is expressed in the isthmus, stays on dorsally and becomes what they refer to as MidO?

See our response to Essential revision #1. We add new data to show the expression domains of *Fgf17* in the IsO and MidO are distinct.

"in turn promotes cell proliferation and neurogenesis through ERK activation in the surrounding cells along the midline". The authors show no proof of this, so it should be proven or said as a suggestion.

We agree that our data are only suggestive. We changed the text as: “we show that these midline cells produce *Fgf17*, which is associated with robust cell proliferation and neurogenesis through ERK activation in the surrounding cells along the midline of the developing cerebellum (Figure 2D-F)”.

Figure 4D. The data presented does not support the conclusion that only one of the subtypes have FoxP1/2 expression, since the IF suggests that most of the PCs express them. Please elaborate more on this. Also more validation of the other markers used to identify the 5 putative PC types is required. With the present data, the presence of these subtypes is not supported.We have provided new IHC data to show PC clusters with distinct combinations of Calb1, Dab1, Foxp1, Foxp2, and Nrgn (Figure 4G).Figure 7. Do any of the transcription factors identified in the different lineages have common upstream regulatory signals?

We believe that this question is beyond the scope of the current study.